# Brain functional-structural gradient coupling reflects development, behavior and genetic influences

Simiao Gao[1,3], Zhiling Gu [1,3], Shengxian Ding [1,3], Gefei Wang [1], Zhengwu Zhang[2], Hongyu Zhao [1] & Yize Zhao [1] ✉

Gradients provide low-dimensional representations of macroscale brain organization, yet how structural-functional gradient coupling develops and relates to behavioral and molecular features remains unclear. Here, we studied structural-functional gradient coupling across multiple metrics and spatial scales using high-resolution structural and functional connectivity from 5343 children in the Adolescent Brain Cognitive Development study and 875 adults from the Human Connectome Project. We find that gradient coupling shows developmental refinement from childhood to adulthood and distinct sex-specific patterns. Gradient coupling metrics are significantly associated with cognitive and mental health measures and enable robust out-of-sample prediction. Heritability analyses reveal that gradient coupling is strongly influenced by genetic factors. Transcriptomic analyses further demonstrate that highly heritable coupling patterns are enriched for genes expressed in deep-layer excitatory neurons. Together, our findings establish structural-functional gradient coupling as a biologically meaningful feature of brain organization that bridges macroscale connectivity, cognition, behavior, and molecular architecture.

Understanding how the anatomical architecture of the brain supports its functional activity is a central question in neuroscience. Structural and functional connectivity (SC and FC) provide complementary, network-level characterizations of the brain structural and functional organization, where SC captures physical pathways of white matter tracts while FC characterizes temporal synchronization of activity between regions across the whole brain. The relationship between SC and FC, known as structure-function (SF) network coupling, reflects how closely brain functional patterns are shaped by anatomical constraints at a large-scale network level[1,2]. Prior work on SF coupling spans infant[3], youth[4], and adult cohorts[5–8]. Importantly, developmental studies also show that SF network coupling increases with age in several cortical territories and follows nonuniform maturational trajectories across the cortex[4,9,10]. Together, these observations suggest that the SC-FC relationship is neither spatially homogeneous nor developmentally static, but instead varies systematically along cortical hierarchies.

Concurrently, recent advances in gradient-based approaches have deepened our understanding of large-scale brain organizations. Rather than characterizing the brain as a set of discrete networks, gradient-based approaches reduce high-dimensional connectivity data into a few continuous axes that capture gradual transitions in the brain organization[11]. Functional connectivity gradients (FCGs) describe smooth spatial variation in FC, revealing continuous transitions from sensory-motor regions toward association cortex, as well as additional axes reflecting other large-scale functional distinctions[12]. Structural connectivity gradients (SCGs) characterize analogous patterns in SC, delineating axes such as somatosensory-medial prefrontal and ventral-dorsal visual transitions[13]. Building on these ideas, we focus on structural-functional gradient coupling (SFGC): the spatial alignment

[1]Department of Biostatistics, Yale University, New Haven, CT, USA. [2]Department of Statistics and Operations Research, University of North Carolina at Chapel Hill, Chapel Hill, NC, USA. [3]These authors contributed equally: Simiao Gao, Zhiling Gu, Shengxian Ding. ✉e-mail: yize.zhao@yale.edu

between SCGs and FCGs, which provides a complementary and more hierarchical perspective on SF coupling. Unlike traditional node-wise SF coupling, which quantifies how similar each region's SC and FC profiles are, SFGC captures how structural and functional hierarchies correspond across the cortex. This distinction is particularly important for neurodevelopment because gradients index large-scale hierarchical organization, whereas node-wise SF coupling reflects only local SC-FC correspondence. Divergent findings between SFGC and node-wise SF coupling, therefore, could highlight distinct multiscale mechanisms of SF organization that cannot be inferred from connectome-level measures alone. Despite its promise, SFGC remains largely understudied. Most research has focused on gradients within a single modality, often limited to FCGs[12], while the cross-modal correspondence between SCGs and FCGs has received limited attention. Prior work has shown that microstructural gradients and FCGs are increasingly dissociated in transmodal cortex[7,14] and that SC-FC correspondence decreases along the functional principal gradient during development[10]; however, no studies have directly examined how SFGC develops over time, to the best of our knowledge. Moreover, the behavioral and genetic relevance of individual differences in SFGC, and whether these relationships differ across development, remain unclear. Filling these gaps could offer critical insight into how the brain integrates structure and function over neurodevelopment, how these interactions support cognition and mental health, and how they are shaped by molecular architectures.

While prior research has suggested that SF coupling may support cognitive performance and clinical outcomes[1,6], the relationship between SFGC and individual differences in cognition or mental health has been underexplored. In particular, the potential of SFGC as a predictive feature for behavioral traits has not been investigated. It remains unclear whether variation in the alignment of structural and functional hierarchies can explain altered cognition and behavior, and if so, which coupling patterns are most informative. Addressing this gap is critical for advancing our understanding of how macroscale brain organization supports complex traits and for identifying potential neuromarkers to inform neurodevelopmental and psychiatric conditions.

Furthermore, both SC and FC are known to be heritable[15,16], and emerging evidence also suggests that SF coupling is shaped by genetic factors[17]. However, it remains unclear whether the alignment between SCGs and FCGs is heritable, which brain systems are most susceptible to genetic influence, or how such effects vary across cortical hierarchies or developmental stages. Meanwhile, transcriptomic atlases such as the Allen Human Brain Atlas (AHBA) now enable researchers to examine whether neuroimaging-derived phenotypes reflect underlying cell-type-specific gene expression patterns[18]. If spatial variations in SFGC correspond to distinct molecular signatures, it would help provide crucial biological context for interpreting this emerging marker for brain organization.

To address these gaps, in this work, leveraging multi-dimensional data from 5343 children in the Adolescent Brain Cognitive Development (ABCD) study and 875 young adults from the Human Connectome Project (HCP), we construct brain SFGC under various metrics and spatial scales across development stages, and comprehensively uncover its correspondence with cognitive and behavioral domains, as well as genetics and molecular architectures. Our findings suggest age-associated changes in SFGC, with observable differences between male and female groups. We further demonstrate that SFGC robustly predicts a wide range of cognitive and mental health traits across cohorts and methods, establishing it as a behaviorally meaningful neuromarker. Extending beyond phenotype, we show that brain SFGC is highly heritable, exhibiting a developmental specificity where genetic influence targets unimodal and transmodal networks in childhood and shifts to preferentially constrain transmodal hubs in adulthood. Finally, we perform imaging transcriptomic analyses to characterize the spatial correspondence between regional heritability maps and cortical gene expression profiles, revealing that heritable SFGC patterns show distinct molecular associations across development: in adults, regions with stronger genetic influence spatially overlap with genes enriched in superficial and intermediate-layer excitatory neurons, whereas in children they exhibit significant correspondence with endothelial and microglial signatures. Together, these findings position SFGC as a biologically grounded, developmentally refined, and behaviorally relevant feature of human brain organization.

## Results

### SFGC reveals developmental refinement across cortical hierarchies

Leveraging multimodal neuroimaging data from two independent cohorts, including children from the ABCD study and young adults from the HCP cohort, we constructed SC and FC for each subject from T1-weighted, diffusion, and functional magnetic resonance imaging (MRI) data. We then derived individual-level cortical SCGs and FCGs, and constructed both macroscale-level and subnetwork-level SFGCs as illustrated in Fig. 1. The detailed data processing, data harmonization and metrics construction procedures are described in the Methods section.

To investigate SFGC and its developmental dynamics, we first examined the architectures of SCGs and FCGs separately. The averaged SC and FC matrices displayed clear network-level structure (Supplementary Fig. 1a), forming the connectivity framework for subsequent gradient and coupling analyses. Applying diffusion embedding to the connectivity affinity matrices yielded a small number of principal gradient axes that summarize large-scale patterns in SC and FC. Visualization of the first two gradients across cohorts was provided in Fig. 2a, b, g, h, illustrating how their spatial layout differed with age. In addition, heatmaps of the False Discovery Rate (FDR)-adjusted p-value ($p$) from 10,000 spin permutations further confirmed that the spatial correspondence between children and adults differed significantly across multiple SCGs and FCGs (Supplementary Fig. 1b, c). In children, the first FCG predominantly captured a sensorimotor-to-visual axis, reflecting the early organization of primary systems. Their second FCG showed strong similarity to the adult primary axis ($r = 0.84$, spin permutation test adjusted $p = 0.03$), indicating the emergence of transmodal association networks. In contrast, adults exhibited a well-defined first FCG extending from primary sensorimotor areas to transmodal hubs, consistent with the mature hierarchical organization of the cortex. SCGs also showed pronounced developmental differences. In children, the first SCG separated the Visual and Somatomotor Networks (VIN and SMN) and the second SCG displayed strong asymmetry in networks across hemispheres (paired t-test $p = 1 \times 10^{-10}$). In adults, the first SCG similarly separated VIN and SMN, while the second SCG did not exhibit significant hemispheric asymmetry (paired t-test $p = 0.52$), displaying a bilaterally symmetric spatial pattern.

The spatial correspondence between SCGs and FCGs reflected a core principle of SF association across cortical hierarchies. To quantify this correspondence, we computed pairwise cosine similarity between the cohort-level SCGs and FCGs, resulting in macroscale SFGC matrices (Fig. 2c, i). Throughout this article, we denoted SFGC between the $i$th FCG and the $j$th SCG as FCG$i$:SCG$j$. Inspection of the SFGC matrices showed distinct correspondence patterns across gradient pairs. In children, a relatively strong coupling (magnitude of SFGC $c = 0.59$) was observed between the first FCG and SCG, suggesting that early functional organization, particularly along the sensorimotor-to-visual axis, remains closely aligned with the underlying structural scaffold. In contrast, the same gradient pair in adults showed much weaker coupling ($c = 0.08$, two-sample Welch's t-test adjusted $p < 10^{-6}$). To formally characterize differences between children and adults, we constructed a heatmap of FDR-corrected p-values for SFGC across all

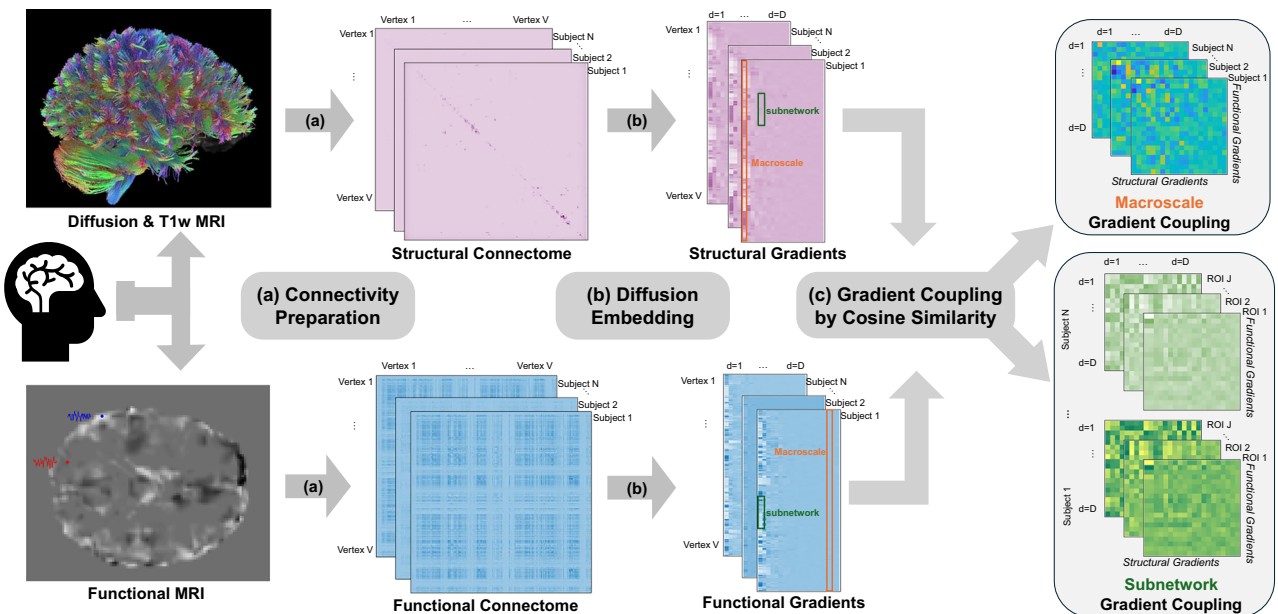

**Fig. 1 | Overview of SFGC construction. a** Preprocessed diffusion MRI, T1-weighted MRI, and resting-state functional MRI were processed using the Surface-Based Connectivity Integration pipeline[53] to generate individual-level structural and functional connectomes. **b** Functional and structural gradients were then extracted from the resulting connectivity matrices using diffusion embedding via the BrainSpace toolbox[57], yielding low-dimensional representations of cortical organization. **c** SFGC was quantified at both the macroscale and subnetwork levels using cosine similarity between corresponding functional and structural gradients. Abbreviation: ROI Region of Interest.

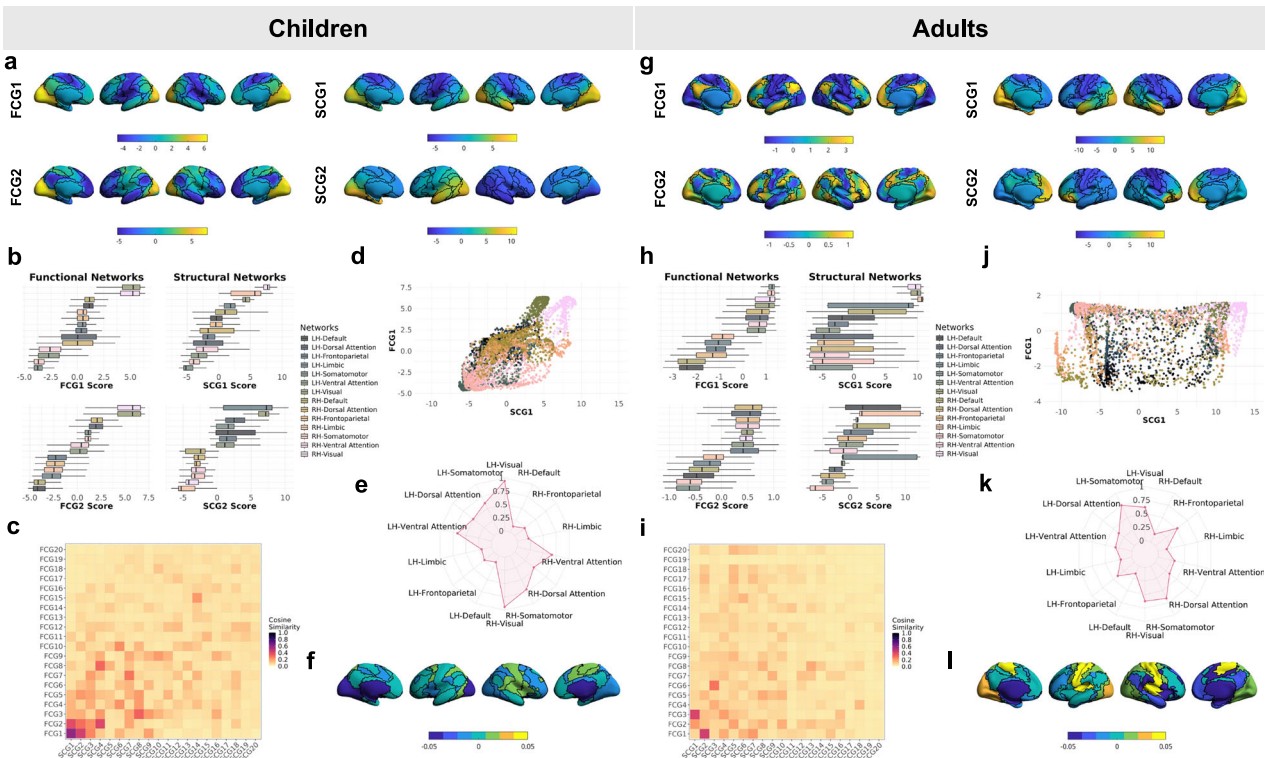

**Fig. 2 | Cohort-level gradients and coupling patterns.** Cohort-level SCGs and FCGs and their couplings at macroscale and subnetwork levels in children (**a**–**f**) and adults (**g**–**l**). **a, g** First and second gradients (FCG1, FCG2, SCG1, SCG2) displayed on the cortical surface with color scales indicating gradient values. **b, h** Distributions of vertex gradient values across the Yeo-7 functional networks averaged from children (*n* = 5343) and adults (*n* = 875). Boxplots summarize the distribution of vertex-level scores within each network, ordered by median value (center line = median; box = interquartile range (IQR); whiskers = 1.5 × IQR). **c, i** Matrices of absolute pairwise cosine similarity between the top 20 SCGs and FCGs. Structural gradients (SCG1 - SCG20) are shown on the x-axis and functional gradients (FCG1 - FCG20) on the y-axis. **d, j** Scatter plots of FCG1 versus SCG1. Color coding matches the networks shown in (**b, h**). **e, k** Radar plots showing the magnitude of SFGC (FCG1:SCG1) within each Yeo-7 network. **f, l** Differences in Yeo-7 subnetwork-level SFGC (FCG1:SCG1) between males and females, plotted as the mean coupling in males minus that in females for each subnetwork. Abbreviations: LH Left Hemisphere, RH Right Hemisphere. Source data are provided as a Source Data file.

structural-functional gradient combinations, which revealed that most pairs exhibited significant cohort differences (Supplementary Fig. 1d).

Having established the macroscale SFGC strength across different gradient pairs and development stages, we next examined how SFGC patterns manifested within individual functional systems, as defined by the Yeo-7 canonical subnetworks[19]. We observed substantial heterogeneity in the gradient alignment at the subnetwork level: some networks exhibited strong coupling even when global alignment was weak, whereas others displayed decoupling despite strong global coupling. To illustrate this coupling heterogeneity, we focused on a representative gradient pair, FCG1:SCG1, which showed relatively high alignment in children but weak alignment in adults (Fig. 2c, i). Notably, strong coupling was consistently observed in primary sensory and motor networks, whereas association networks such as Default Mode Network (DMN) showed greater decoupling (Fig. 2d, j). To further assess these network-specific patterns, we visualized coupling strength across networks in each hemisphere using radar plots (Fig. 2e, k). The results highlighted relatively high and consistent gradient alignment in unimodal systems across developmental stages, in contrast to more variable couplings observed in other networks. Specifically, the Ventral Attention Network (VAN) was more pronounced in children, while Frontoparietal Network (FPN) was relatively high in adults. Together, these findings established that both macroscale- and subnetwork-level SFGC represent meaningful levels of organization, and that examining both scales provides a more complete characterization of SF correspondence. We therefore included both metrics in downstream analyses to assess their associations with behavioral, demographic, and genetic factors.

### SFGC and mental health associations by age and sex

Associations between SFGC and a broad range of behavioral and demographic outcomes were assessed at both macroscale and subnetwork levels. To ensure robust and generalizable estimates, we implemented two predictive modeling approaches: kernel ridge regression (KRR)[20] and multilayer perceptrons (MLP)[21], with 100 random train-test splits. Detailed procedures for model fitting and evaluation are provided in the Methods section.

We examined associations between SFGC and mental health outcomes, motivated by prior evidence linking SF coupling to psychiatric vulnerability[22]. Our results revealed modest average out-of-sample associations between SFGC and mental health outcomes. In particular, the association between SFGC and mental health outcomes varied by spatial scale and age cohort: in children, macroscale-/subnetwork-level SFGC yielded average $r = 0.09/0.11$ with mental health outcomes; whereas in adults, SFGC exhibited a slightly weaker association with average $r = 0.04/0.11$ (Fig. 3a). Furthermore, in both cohorts, the strongest association was observed for rule-breaking (RuleBreak) symptoms with subnetwork SFGC ($r = 0.145$ in children, $r = 0.144$ in adults). The FDR-corrected paired t-tests of the difference between macroscale- and subnetwork-level SFGC in Fig. 3a demonstrated overall subnetwork SFGC's stronger association with mental health outcomes, e.g., aggressive (Aggressive), withdrawn/depressed (With/Dep), RuleBreak, and externalizing (External) symptoms. This divergence suggested that the spatial scale at which SFGC was behaviorally relevant may shift across development.

Beyond mental health, SFGC also showed predictive utility for sex classification, with area under the curve (AUC) values of 0.77/0.83 in children and 0.79/0.87 in adults (Fig. 3a) for macroscale-/subnetwork-level SFGC. Visual inspection of SFGC topographies revealed distinct spatial patterns between males and females (FCG1:SCG1, Fig. 2f, l). Specifically, sex differences were more pronounced in unimodal regions in adults, whereas in children, they appeared in a different cortical pattern, suggesting that SFGC may capture a sex-dimorphic dimension of brain architecture.

To formally assess cohort-related differences, we conducted FDR-corrected two-sample Welch's t-tests comparing SFGC-behavior association strengths between children and adults. As shown in Fig. 3b, SFGC was more strongly associated with sex and age in adults, while SFGC was in general more associated with mental health outcomes (Aggressive, External) in children (adjusted $p < 0.01$). Given that SFGC underwent extensive harmonization, the persistence of this pattern likely reflected genuine developmental shifts in brain architecture.

To explore how these sex and cohort effects interact in shaping the mental health relevance of SFGC, we further stratified the cohorts by sex and re-estimated SFGC-outcome associations (Fig. 3c). In children, males showed greater associations for External, internalizing (Internal), Aggressive, and anxiety/depression (Anx/Dep) for FDR-corrected Welch t-tests. In adults, however, females exhibited a stronger association of subnetwork SFGC with Aggressive, while males maintained the stronger macroscale SFGC association with External, RuleBreak and Internal. These findings suggested the mental health correlates of SFGC were modulated by both development and sex.

### Cognitive relevance of SFGC varies across development and sex

Previous studies reported significantly altered SC and FC between hippocampal subregions, implicated in cognition and emotion, and cortical areas in individuals with cognitive impairment[23,24]. Furthermore, while SF coupling was found to be largely preserved at the whole-brain level in mild cognitive impairment, it was selectively increased in the Dorsal Attention Network (DAN) and decreased in the VAN[25]. These findings suggest that SFGC may reflect compensatory or pathological reorganization and could serve as a predictor of cognitive outcomes.

Building on this, we evaluated the relationship between SFGC and cognitive function across development. SFGC showed broadly similar association patterns, with subnetwork-level coupling showing slightly stronger associations with most cognitive outcomes (Fig. 3a). Specifically, the FDR-corrected paired t-tests for the difference between macroscale- and subnetwork-level SFGC in their association with cognition demonstrated that subnetwork-level SFGC associated with multiple outcomes better than macroscale-level SFGC in both cohorts (adjusted $p < 0.001$), such as Picture Sequence Memory (PicSeq), Picture Vocabulary (PicVocab), and Total Cognition Composite (CogTotal). Overall, this showcased the additional information embedded in subnetwork SFGC and its significance in downstream analysis.

In addition, the strength of the association between SFGC and cognitive performance was greater in children on average (macroscale/subnetwork: $r = 0.26/0.30$), particularly for integrative cognitive scores such as CogTotal (macroscale/subnetwork: $r = 0.38/0.44$). Lower association between cognitive outcomes and SFGC was observed in the adult cohort (macroscale/subnetwork: $r = 0.18/0.23$), among them the highest association was observed for Crystallized Composite score (CogCrystal, macroscale/subnetwork: $r = 0.28/0.38$). The two-sample Welch's t-tests in Fig. 3b further confirmed that SFGC was more predictive of cognitive outcomes in children, including List Sorting Working Memory (ListSort), Fluid Cognition Composite (CogFluid), Dimensional Change Card Sort (CardSort), CogTotal, PicVocab, and CogCrystal (adjusted $p < 0.001$), underscoring the greater behavioral relevance of gradient alignment during this critical developmental stage.

Having observed sex differences in the associations between SFGC and mental health, we next examined whether similar sex-dependent patterns exist for cognitive outcomes across development. As shown in Fig. 3c, in children, females exhibited stronger SFGC-cognition associations for PicSeq, and males showed stronger association Flanker Inhibitory Control and Attention (Flanker) and CardSort (adjusted $p < 0.001$). In contrast, in adults, females showed a stronger association with Flanker and CardSort (adjusted $p < 0.05$), while males showed a stronger associations with PicSeq (macroscale: adjusted $p < 0.001$). In both cohorts, Oral Reading Recognition (ReadEng) remained more strongly linked to SFGC in females; and CogFluid and Pattern Comparison (ProcSpeed) were more strongly

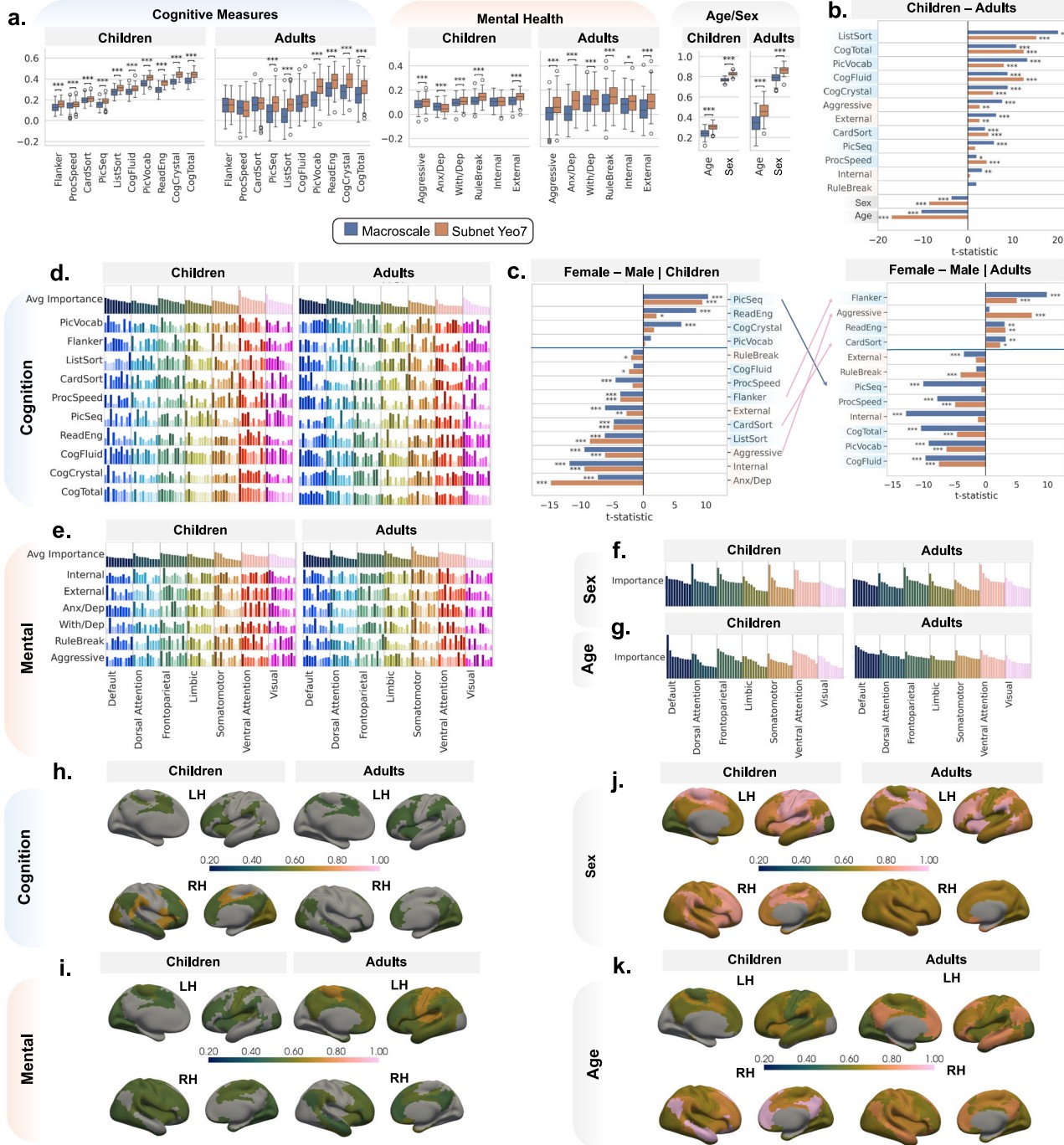

**Fig. 3 | The association between SFGCs and behavioral outcomes and demographics. a** Box plots illustrating SFGC-behavior associations at the macroscale and Yeo-7 subnetwork levels. Correlation coefficients were shown for continuous measures (cognitive, mental health, and age), and AUC values were displayed for sex. Box plots indicate median (middle line), 25th, 75th percentile (box) and 5th and 95th percentile (whiskers) as well as outliers (single points). Asterisks denoted significance level in the difference between macroscale and subnetwork SFGC-behavior associations using FDR-corrected two-sided paired t-test (*: $p < 0.05$, **: $p < 0.01$, ***: $p < 0.001$) using $n = 100$ splitting replicates. **b**, **c** Differences in SFGC-behavior associations across cohorts and sexes. Asterisks denoted significance level in the cohort and sex difference in SFGC-behavior associations using FDR-corrected two-sided Welch's t-test with unequal variance using $n = 100$ splitting replicates. **d–g** Feature importance of top subnetwork-level SFGC metrics for predicting individual behavioral outcomes (lower panels) and domain-averaged importance (top panel). Taller bars indicated greater predictive importance. **h–k** The highest averaged feature importance for SFGC within each subnetwork in predicting domain outcomes. Abbreviations: LH Left Hemisphere; RH Right Hemisphere. For other abbreviations for the behavioral measures, please refer to Supplementary Table 2. Source data are provided as a Source Data file. Exact p-values are reported in Supplementary Data file.

linked to SFGC in males. This developmental reversal for PicSeq, Flanker and CardSort may reflect sex-specific patterns of brain maturation. Together, these findings pointed to a dynamic interaction between SFGC and cognitive functions, shaped by both developmental stage and sex.

## Key SFGCs exhibit a shift across developmental stages
To identify the neural circuits underlying associations between SFGC and behavioral and demographic outcomes, we analyzed the feature importance of subnetwork-level SFGC measures, defined as normalized primal coefficients for KRR and normalized saliency maps for

MLP, see details in Methods section. We report the maximum feature importance (FI) within each subnetwork in the following. A clear cohort-related redistribution of SFGC emerged in associations with sex (Fig. 3f, j). In children, predictive SFGC features for sex were primarily localized in the SMN and DAN (FI = 0.94, 0.95), whereas in adults, these features shifted toward the FPN and VAN (FI = 0.91, 0.96), suggesting a maturation-related transition from sensory-driven (unimodal) to integrative (transmodal) functional systems.

For cognitive outcomes (Fig. 3d, h), we also observed a shift in the spatial distribution of predictive subnetwork features. In children, predictive SFGC located primarily in VAN and VIN (FI = 0.70, 0.62). In contrast, adults showed numerically decreased involvement of VAN (FI = 0.51). For SFGC-age association (Fig. 3g, k), children appeared to rely more heavily on the DMN, FPN and VAN (FI = 0.99, 0.74, 0.71), while in adults, predictive contributions tend to come increasingly from the SMN, DAN and VAN (FI = 0.72, 0.72, 0.72). Additionally, for SFGC-mental health association (Fig. 3e, i), we observed a numerically higher involvement of SMN in adults (FI = 0.71) than in children (FI = 0.56), while a numerically lower involvement of VIN was observed in adults (FI = 0.44) than in children (FI = 0.53).

Moreover, a clear developmental distinction emerged in Fig. 3a, where SFGC-behavior associations showed lower inter-individual variability in children than in adults across all behavioral domains. Taken as a whole, these results suggested that SFGC became increasingly differentiated and individualized with age, offering a potential neural substrate for behavioral diversity in adulthood. Additionally, as shown in Supplementary Fig. 3, we observed consistent overall patterns in the FI maps, supporting the robustness of our analysis and the reliability of the proposed FI metrics.

## SFGC exhibits distinct and hierarchical genetic contributions

We further investigated how genetic factors shape SFGC across developmental stages by estimating the heritability of each coupling metric. Specifically, we computed SFGC heritability using both the Yeo-7 canonical subnetworks and the D-K atlas[26] at a finer resolution, with the latter enabling a more precise characterization of the spatial distribution of heritability across the brain and a direct comparison of these patterns with the spatial organization of SFGC itself. As shown in Fig. 4a, b, j, k, the average SFGC strength matrices for both children and adults exhibited a clear hierarchical organization across gradient orders: couplings involving the principal (lower-order) SCGs and FCGs showed the strongest alignment, whereas those involving higher-order gradients were markedly weaker. Following this same hierarchy, the heritability of these SFGC metrics revealed an even more explicit and systematic decline across gradient orders in both cohorts and parcellation schemes (Fig. 4c, d, l, m). Couplings anchored on the principal gradients showed the strongest genetic influence, whereas couplings involving progressively higher-order gradients showed substantially lower heritability. This pattern was particularly pronounced under the D-K atlas, likely due to its finer spatial resolution that enhanced the capture of localized, region-specific genetic signals. Importantly, this hierarchical decline in heritability was not simply a byproduct of weaker coupling strength at higher gradient orders. Rather, it reflected an underlying gradient-based genetic hierarchy, suggesting that lower-order SFGCs, presumed to capture core axes of SF brain organization, were more strongly constrained by genetic factors.

The magnitude of SFGC heritability was broadly consistent across parcellations and cohorts, with averages that were slightly higher in children than in adults (Yeo-7: $h^2$ = 19.45% versus 18.32%; D-K: $h^2$ = 17.42% versus 16.48%). The highest heritability estimates for individual subnetwork-level SFGC reached 61.99% in adults and 60.02% in children. At the macroscale-level SFGC, the mean heritability increased further to 24.13%, with maximum values up to 61.56%. To contextualize these estimates, we compared them to the heritability of unimodal

gradients (Supplementary Fig. 4). While FCGs and SCGs showed moderate to strong heritability, SFGC exhibited mean heritability comparable to these unimodal gradients but reached higher maximal values. Notably, paired comparisons after multiplicity correction (Supplementary Table 1) revealed that SFGC displayed significantly higher heritability than corresponding FCGs in both cohorts, and significantly higher heritability than SCGs in children but lower heritability than SCGs in adults. These findings indicated that SFGC captures highly heritable multimodal axes of cortical organization whose genetic influence differs systematically from unimodal gradients and shifts across development.

## SFGC reveals regional and developmental genetic topographies

Building on the hierarchical heritability patterns across gradient indices, we further examined the spatial topography of SFGC heritability across the cortex. We systematically quantified heritability for all 400 coupling pairs, characterizing both global average patterns and the subnetwork-level distribution of genetic influence. In the following, we highlight the FCG1:SCG1 coupling as a representative example, as it consistently exhibited strong heritability in both children and adults. At the Yeo-7 subnetwork level (Fig. 4e, n, g, p), the spatial pattern of FCG1:SCG1 heritability closely matched the generalized heritability distributions. In adults, both the surface map of FCG1:SCG1 heritability and the subnetwork-level heritability distributions highlighted the DMN and FPN as heritability hotspots, with significantly higher values in the left hemisphere (LH; DMN: adjusted $p$ = 0.006; FPN: adjusted $p$ = 0.010), suggesting strong and lateralized genetic constraint on transmodal systems during maturity. In children, peak heritability was localized primarily within the VIN and DMN. Only the VAN showed significant hemispheric asymmetry (adjusted $p$ = 0.037), again favoring the LH. Across cohorts, the VIN and DMN consistently showed high median heritability, while the Limbic Network ranked lowest. These findings collectively indicated a developmental progression from broad genetic constraints in unimodal and attentional systems in childhood to strongly lateralized transmodal hubs in adulthood.

Regional analyses using the D-K atlas (Fig. 4f, o, i, r) refined these patterns. In adults, FCG1:SCG1 heritability was highest in the superior frontal gyrus, precuneus, and inferior parietal cortex, and the full distribution across coupling traits confirmed peak densities in bilateral precuneus, a pivotal DMN hub, alongside the LH inferior parietal and right middle temporal cortices. In children, heritability was strongest in occipital regions and the precuneus, with the bilateral precuneus, right hemisphere (RH) lingual gyrus, and LH lateral occipital cortex showing maximal density. These findings revealed a shift from occipital-precuneus dominance in children to precuneus-frontoparietal dominance in adults, with the precuneus emerging as a developmentally stable genetic hotspot.

Finally, hemispheric correlations in regional heritability (Fig. 4h, q) showed strong bilateral similarity in adults ($r$ = 0.69, $p$ = 6.97 × 10⁻⁶) and moderate similarity in children ($r$ = 0.56, $p$ = 5.77 × 10⁻⁴). Global lateralization was nonsignificant in both cohorts (adults: $p$ = 0.798; children: $p$ = 0.672), yet regional asymmetries were evident. In children, higher RH heritability appeared in the superior frontal and posterior cingulate gyri; in adults, the LH superior frontal and RH isthmus cingulate showed prominent asymmetry. Thus, while overall genetic constraint remained bilaterally balanced, region-specific asymmetries shifted developmentally, highlighting dynamic and genetically regulated functional lateralization across development.

## Transcriptomics highlights cellular bases of SFGC heritability

To investigate whether regional variation in SFGC heritability exhibits a structured spatial correspondence with cortical gene expression patterns, we performed imaging transcriptomic analyses by integrating regional heritability estimates with developmentally matched bulk and single-cell gene expression datasets (Fig. 5a, b). Using spatially

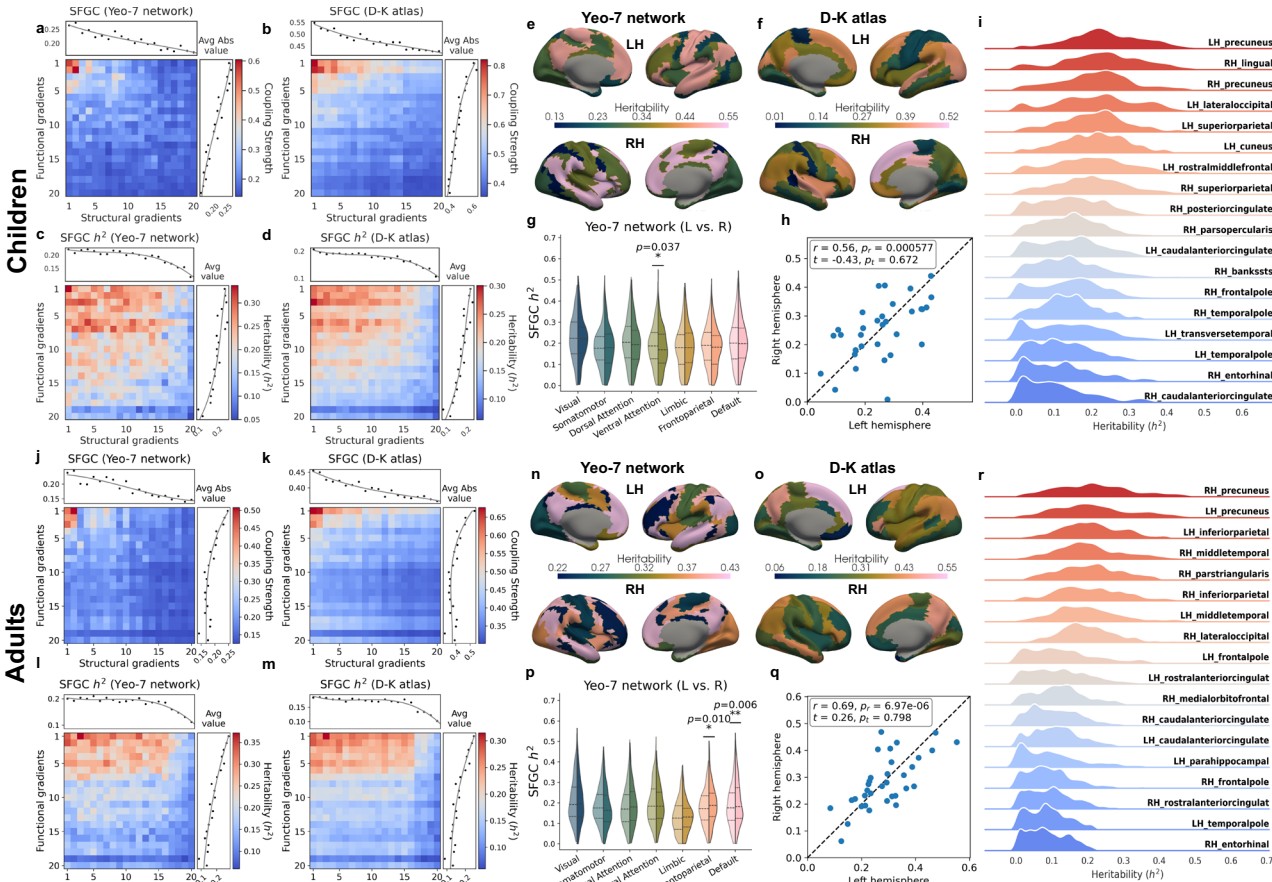

**Fig. 4 | Heritability analyses of SFGC.** Subnetwork-level SFGC and its heritability in the two cohorts, averaged across the Yeo-7 subnetworks (**a, c, j, l**) and D-K atlas (**b, d, k, m**) respectively. **e, n** Heritability estimates of Yeo-7 subnetwork-level coupling between FCG1 and SCG1. **f, o** Heritability estimates of D-K atlas-level coupling between FCG1 and SCG1. **g, p** Distribution of heritability for Yeo-7 sub-network-level SFGC, separated by hemispheres. Asterisks denote significance based on two-sided Bonferroni-corrected paired t-tests

(*: $p < 0.05$, **: $p < 0.01$, ***: $p < 0.001$). **h, q** Heritability of the D-K atlas-level SFGC between FCG1 and SCG1, grouped by hemisphere, along with corresponding Pearson correlations ($r$), two-sided t-statistics from paired t-tests, and p-values ($p_r$ and $p_t$ respectively). **i, r** Distribution of heritability estimates for D-K atlas-level SFGC. Abbreviations: LH Left Hemisphere; RH Right Hemisphere; L Left; R Right. Source data are provided as a Source Data file.

adjusted gene-heritability correlations and cell-type-specific gene set enrichment, we identified distinct transcriptomic profiles associated with the representative FCG1:SCG1 coupling across development (Fig. 5d, e).

In adults, regional SFGC heritability showed significant positive spatial correspondence with gene sets associated with excitatory neurons enriched in superficial and intermediate cortical layers (Excitatory neuron 3: adjusted $p = 0.009$; Excitatory neuron 4: adjusted $p = 0.042$; Excitatory neuron 5: adjusted $p = 0.022$; Excitatory neuron 8: adjusted $p = 0.047$). These excitatory classes are crucial for intracortical communication and long-range integration, properties consistent with the transmodal regions exhibiting the strongest heritability. Conversely, significant negative correspondence was observed with Microglia (adjusted $p = 0.015$) and Astrocytes (adjusted $p = 0.046$). In children, SFGC heritability showed significant positive spatial correspondence with gene sets associated with Endothelial (adjusted $p = 0.015$) and Microglia (adjusted $p = 0.0004$), pointing to a stronger contribution of vascular and immune-related pathways during earlier developmental stages. Together, these findings indicated developmentally distinct molecular patterns underlying the spatial distribution of SFGC heritability.

We also examined individual genes exhibiting both high cell-type specificity and consistent spatial association with SFGC heritability across cohorts. We focused on coding genes corresponding to two excitatory neuron-preferential RNA-binding proteins, *CPEB3* and

*NOVA1* (Fig. 5c), known master regulators of synaptic function. In adults, both *CPEB3* ($r = 0.33$, adjusted $p = 0.0345$) and *NOVA1* ($r = 0.33$, adjusted $p = 0.0267$) showed significant positive correlations with the regional heritability of SFGC (Fig. 5g). Similar associations were observed in children (Fig. 5f; *CPEB3*: $r = 0.55$, adjusted $p = 0.0347$; *NOVA1*: $r = 0.53$, adjusted $p = 0.0492$). While broader excitatory gene sets were not enriched in children, these genes provided isolated examples of spatial correspondence that is conserved across development.

## Discussion

This work provides a comprehensive characterization of brain SFGC across late childhood and young adulthood and demonstrates how SFGC relates to individual differences in brain organization and behavior. Prior research primarily focused on FC-SC correspondence[3–8] and single-modal gradients[12,27], with relatively limited emphasis on the cross-modal alignment between SCGs and FCGs, and no systematic examination of its alterations over developmental stages, behavioral relevance, or genetic architecture. Here, we introduced SFGC as a neurobiologically grounded phenotype that quantifies the spatial alignment at various scales between structural and functional hierarchies in the human brain. Using multimodal neuroimaging data from two large-scale independent cohorts spanning childhood to adulthood, we demonstrated that SFGC captured key dimensions of neurodevelopmental reorganization and linked

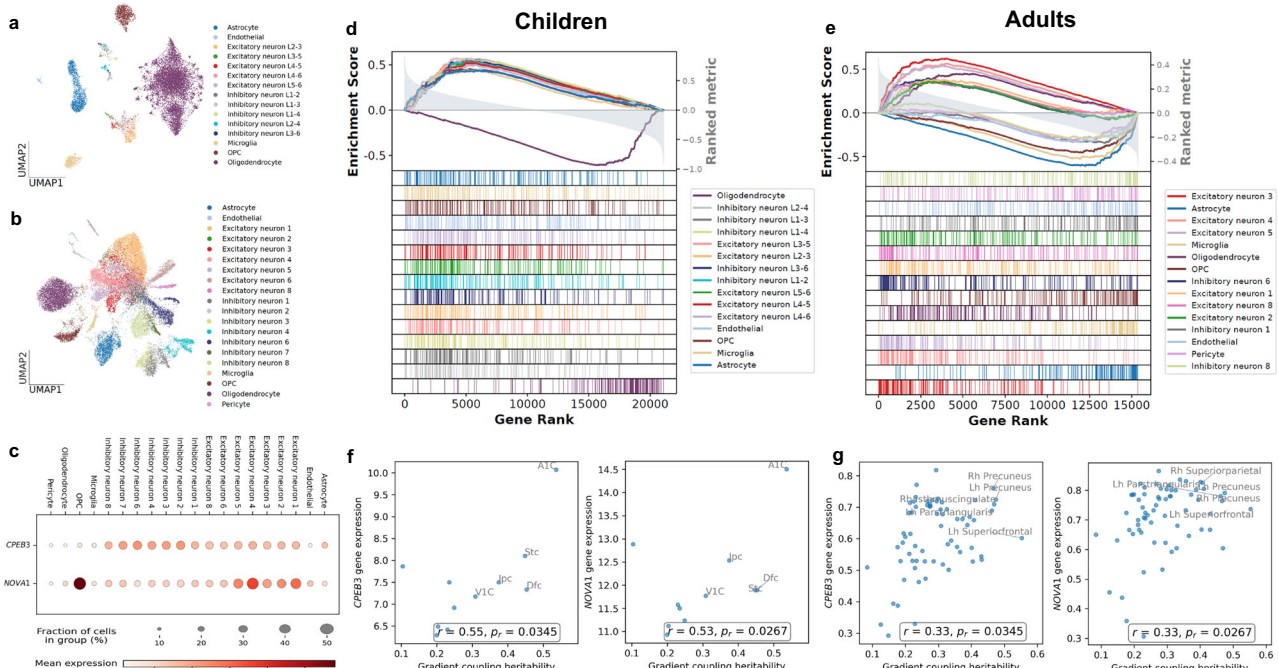

**Fig. 5 | Imaging transcriptomics analyses of SFGC. a, b** UMAP visualization[69] of the scRNA-seq data used for the children cohort[64], and the young adult cohort[61], with cells colored by annotated cell types. **c** Dot plot of representative genes *CPEB3*, and *NOVA1*, showing their mean expression and fraction of cells expressing them across the major cell types in the cortex. **d, e** GSEA results for the heritability of region-wise coupling between FCG1 and SCG1 in the two cohorts. **f, g** Scatter plots showing associations between the heritability of SFGC (FCG1:SCG1) and expression levels of the genes in **c**, along with corresponding Spearman's correlation coefficients ($r$) and two-sided p-values adjusted for spatial autocorrelation ($p_r$). Source data are provided as a Source Data file.

macroscale brain architecture to cognition, behavior, and molecular mechanisms.

One of our central findings was the developmental difference in SFGC from childhood to adulthood. In children, major functional organization was closely aligned with the principal structural axes, whereas in adults the pattern of alignment was more distributed across gradients, reflecting a more differentiated organization of SF relationships. This is consistent with prior evidence for age-related, spatially heterogeneous reorganization of SF coupling across cortical systems[9]. A related developmental feature of SCGs was the dissociation between large-scale spatial organization and local dispersion of gradient values. In adults, principal structural gradients exhibited clearer large-scale spatial organization, including more pronounced bilateral symmetry, while at the same time showing greater dispersion of gradient values within specific subnetworks. Importantly, these observations reflected distinct aspects of gradient structure: large-scale spatial organization captures the gradient layout across the cortex, whereas local dispersion reflects heterogeneity in gradient expression across vertices within subnetworks. From a developmental perspective, this apparent dissociation suggested that maturation may support more stable global organization of structural gradients while allowing increased heterogeneity in how individual vertices express these gradients.

Our subnetwork analyses further demonstrated that SFGC was not uniformly distributed across the cortex. For example, FCG1:SCG1 coupling was consistently stronger in primary sensory and motor systems and weaker in the DMN. This pattern aligned with prior observations that alignment between SCGs and FCGs was highest in unimodal cortices and reduced in transmodal association regions[7,14]. Developmentally, coupling profiles showed network-specific differences rather than uniform shifts across the cortex. These findings indicated that developmental effects on gradient-level correspondence were selective across networks, with some systems showing stronger coupling (VAN in FCG1:SCG1) in childhood and others (FPN in

FCG1:SCG1) exhibiting greater correspondence in adulthood. They highlighted an important distinction: hierarchical gradient alignment may decrease as cortical systems diversify, while subnetwork-level SF correspondence can become more differentiated across gradients. Importantly, these network-specific interpretations complemented reports such as increasing regional SF coupling within the association cortex during childhood and adolescence[4,9,10].

SFGC also showed systematic variation in behavioral relevance across development and sex. SFGC showed stronger associations with mental health and cognitive outcomes in children than in adults, suggesting that early development may be a period of greater vulnerability to mental and cognitive disorders[28,29]. This finding indicated that SFGC may hold value for early risk assessment in pediatric populations[30]. The stronger predictive value of subnetwork SFGC suggested that localized differences in SFGC carry meaningful information about mental health variation. SFGC provided further evidence for sex-dimorphic aspects of brain organization: prior studies have reported sex differences in large-scale network development[31,32], and our findings extended this work by showing that both the strength and spatial distribution of SFGC-symptom associations differ between males and females. For cognition, subnetwork-level coupling again outperformed macroscale-level coupling, indicating that cognitive variation was more strongly linked to localized SFGC. Developmentally, associations were stronger in children, particularly for composite cognitive measures, consistent with the view that SF alignment played a more prominent role during late childhood and early adolescence, when large-scale networks undergo substantial refinement[33,34]. Sex differences further highlighted stage-specific and domain-specific variability in how SFGC relates to cognition, showing opposite patterns across cohorts in tasks such as PicSeq, Flanker, and CardSort[35].

Our FI analyses revealed a systematic developmental shift in the networks most strongly linked to SFGC. Predictive SFGC for sex in children were primarily anchored in unimodal systems, consistent with

evidence that gradient-based organization was more localized in early-developing sensory cortices[27]. In adults, predictive features for sex shifted toward transmodal systems, aligning with findings that attention and association systems increasingly scaffold cognitive abilities with maturation[36,37]. For mental health associations, stronger SMN involvement and reduced VIN involvement in adults aligned with emerging evidence implicating these circuits in psychopathology[38]. Greater interindividual variability in adults further suggested that SFGC became more differentiated with age, consistent with increasing neural specialization later in development.

The heritability analyses revealed that SFGC was shaped by a hierarchical genetic architecture that reorganized across development. Lower-order gradient pairs were consistently the most heritable in both cohorts, indicating that genetic factors preferentially stabilized core axes that anchored large-scale cortical organization[39,40]. A clear developmental shift was observed: in childhood, genetic effects were strongest in unimodal visual and attentional systems with limited hemispheric specialization, whereas in adulthood, genetic influence became increasingly concentrated and lateralized within transmodal hubs, particularly the DMN and FPN. This shift from occipital-precuneus dominance in children to precuneus-frontoparietal dominance in adults highlighted the precuneus as a developmentally stable genetic hotspot[41]. These results indicated that genetic constraints on SFGC became progressively more focused, hierarchical, and lateralized from childhood to adulthood.

Transcriptomic analyses reinforced this developmental transition by linking heritable SFGC patterns to age-specific cellular substrates. In children, regional heritability was associated with genes expressed by endothelial and vascular-associated populations, consistent with the central role of neurovascular signaling and angiogenesis in early cortical development[42,43]. In adults, enrichment shifted toward multiple classes of intracortical excitatory neurons, aligning with prolonged refinement of synaptic and projection processes supporting higher-order specialization[44,45]. Furthermore, the reversal in microglial associations, positive in children but negative in adults, suggested a developmental transition from early synaptic sculpting to later immune-metabolic roles[46,47]. Collectively, these results outlined a multiscale account in which genetic influences shift from broad, vascular-linked constraints in childhood toward more selective, excitatory-neuron-driven control in adulthood.

While this study offers several insights, it also points to several directions for future research. First, although this study compared children and adults as two broad developmental cohorts, the ABCD baseline sample spans a relatively narrow age range (9–10 years), which constrained the ability to resolve more fine-grained developmental changes in SFGC within childhood. Future work will extend this framework to longitudinal follow-up waves of the ABCD study and to the HCP Development (HCP-D) dataset to capture more continuous maturational trajectories and provide a finer characterization of how SF alignment changes across development. Second, while our use of cosine similarity between low-dimensional gradients under different scales provides an interpretable metric of gradient alignment, it may not capture the potential nonlinear relationship between gradients. Extending this framework to establish more rigorous manifold-based representations for coupling metrics, as well as to incorporate temporal fluctuations within fMRI time series and task-based modulations could yield a more comprehensive understanding of the dynamics and state-specific properties of SFGC.

## Methods

### Ethical compliance and data use
All research described in this study complies with all relevant ethical regulations. This study involves the secondary analysis of data from the ABCD Study and the HCP. The original ABCD Study protocols were approved by a centralized Institutional Review Board (IRB) at the University of California, San Diego, as well as the local IRBs at each of the 21 data collection sites. The HCP study protocols were approved by the IRB at Washington University in St. Louis. All participants or their legal guardians provided written informed consent at the time of original data collection. This study was conducted in accordance with the data use agreements of the ABCD Study and the HCP.

### ABCD study
The ABCD Study, funded by the National Institutes of Health, is a large-scale longitudinal research initiative investigating brain development and child health in approximately 11,880 children, recruited at ages 9–10 from 21 sites across the United States[33]. Employing comprehensive assessments, including advanced neuroimaging, genetic analysis, cognitive testing, mental health evaluations, and detailed environmental measures, the study tracks participants from childhood through adolescence into young adulthood. Its primary goals are to understand developmental trajectories, examine how genetic, behavioral, and environmental factors influence cognitive, emotional, social, and physical growth, and identify early indicators of mental health conditions to inform effective intervention strategies.

The neuroimaging and behavioral data used in this study were obtained from the NIMH Data Archive (NDA, https://nda.nih.gov/abcd). After rigorous quality control and preprocessing of the neuroimaging data, and exclusion of participants with incomplete covariate information, the final analytic sample consisted of 5343 subjects (age: mean = 9.97 years, range = 9.00–10.92 years; sex: 48.1% male). For the heritability analyses, we restricted the sample to 975 subjects drawn from 483 families contributing related pairs, which includes 120 monozygotic (MZ) twin pairs, 176 dizygotic (DZ) twin pairs, and 223 full sibling pairs.

### HCP
The HCP is a major initiative aimed at comprehensively mapping neural pathways to elucidate the brain's complex wiring architecture. A key component of this effort is the HCP Young Adult (HCP-YA) cohort (https://www.humanconnectome.org/study/hcp-young-adult/data-releases), which focuses on individuals aged 22 to 35. This life stage is characterized by relative brain maturity and stability, making it ideal for investigating normative brain structure and function. Furthermore, we utilized the HCP-D cohort (https://www.humanconnectome.org/study/hcp-lifespan-development/overview), part of the Lifespan HCP consortia, as a strategic bridge to facilitate harmonization between the childhood and adult cohorts. The HCP has established rigorous standards for data acquisition and preprocessing, providing an unprecedented view of structural and functional connectivity. Beyond neuroimaging, participants complete a battery of behavioral and cognitive assessments, enabling the exploration of brain-behavior relationships. Neuroimaging data and most behavioral measures are publicly available at https://db.humanconnectome.org, subject to the HCP Open Access Data Use Terms.

Following quality control and data aggregation, our final analytic HCP-YA sample comprised 875 participants (age: mean = 28.62 years, range = 22–37 years; sex: 46.74% male) from 407 families, including 104 MZ twin pairs, 54 DZ twin pairs, 478 full sibling pairs, 25 half sibling pairs, and 106 singletons. The final analytic sample for HCP-D involved 93 subjects (age: mean = 10.01 years, range = 9.00–10.92 years; sex: 38.7% male).

### Behavioral outcomes
For both children and young adults, we included the NIH Toolbox Cognition Battery, a standardized and developmentally appropriate assessment recommended for individuals aged 7 and older[48]. This battery spanned multiple cognitive domains, including executive function, episodic memory, language, working memory, processing speed, and sustained attention. It also included composite indices such

as the Total, Fluid, and Crystallized Cognition scores, which provided aggregate measures of global, reasoning-based, and knowledge-based abilities, respectively. In our analyses, we included all seven individual task scores and the three composite scores to comprehensively assess the association between cognitive performance and the proposed coupling metric.

For mental health, we used instruments from the Achenbach System of Empirically Based Assessment: the Child Behavior Checklist (CBCL) for children and adolescents (ages 6–18), and the Adult Self-Report (ASR) for adults (ages 18–59). These parallel tools offered dimensional measures of behavioral and emotional functioning based on caregiver report (CBCL) and self-report (ASR), respectively. Specifically, we examined the broadband internalizing and externalizing scores, as well as key syndrome subscales including anxious/depressed, withdrawn/depressed, rule-breaking behavior, and aggressive behavior. A complete list of all cognitive and mental health measures used in our analyses, along with their corresponding variable codes, is provided in Supplementary Table 2. In addition, we summarized demographic variables as well as cognitive and mental health measures stratified by sex and cohort in the Supplementary Table 3.

## Imaging processing and connectivity construction

The ABCD scan session followed a fixed sequence, beginning with a brief localizer for head alignment, followed by the acquisition of high-resolution 3D T1-weighted structural images, two runs of eyes-open resting-state fMRI, diffusion-weighted imaging, and 3D T2-weighted structural scans[33]. Depending on real-time motion monitoring, one or two additional resting-state fMRI runs were acquired to optimize data quality. The session concluded with task-based fMRI paradigms, including the Monetary Incentive Delay, Stop-Signal, and emotional n-back tasks, which probed reward processing, inhibitory control, and working memory. All data were collected using harmonized protocols across Siemens Prisma, GE 750, and Philips platforms at 21 U.S. sites, and underwent centralized quality control and Brain Imaging Data Structure-compliant preprocessing. Quality control (QC) encompassed both automated and manual processes. Automated QC quantified modality-specific metrics including SNR, temporal SNR, head motion (mean FD), and dMRI-specific artifacts such as dark slices identified via tensor-fit residuals. Trained technicians then visually inspected all modalities and derived images using standardized montages, with datasets showing severe motion or imaging artifacts flagged and excluded from further processing. Furthermore, structural, diffusion, and functional MRI data were preprocessed using the standardized ABCD pipeline. For structural T1w/T2w images, preprocessing included gradient nonlinearity correction, cross-modal registration using mutual information, and advanced bias-field correction to mitigate steep receive-coil intensity variations that can affect cortical surface reconstruction. Diffusion MRI data underwent iterative model-based eddy current correction with robust dark-slice censoring, rigid-body motion correction with rotation-adjusted gradient tables, B0 distortion correction using reversed phase-encode b=0 pairs, gradient nonlinearity correction, and registration to T1w space before resampling to 1.7-mm isotropic resolution. Functional MRI preprocessing included AFNI-based head-motion correction, B0 distortion correction using spin-echo field maps, gradient nonlinearity correction, between-scan rigid alignment to a mid-session reference scan, and registration to structural images, with final data retained in native space at 2.4-mm isotropic resolution. More details can be found elsewhere[49].

The HCP-YA protocol was distributed across four approximately 1-hour sessions over two days, each beginning with a rapid localizer to ensure precise head positioning. Day 1 combined ultra-high-resolution structural imaging (T1- and T2-weighted) with alternating blocks of resting-state and task-evoked fMRI. Day 2 emphasized multi-shell diffusion acquisitions and included additional resting-state and task fMRI runs to enhance test-retest reliability[50]. All scans were performed on a customized Siemens Connectome Skyra with enhanced gradients and multiband EPI sequences. Data were processed through six of the HCP Minimal Preprocessing Pipelines, which included three structural pipelines (PreFreeSurfer, FreeSurfer, PostFreeSurfer), two functional pipelines (fMRIVolume and fMRISurface), and a diffusion pipeline[51]. The structural pipelines corrected spatial distortions, aligned T1w and T2w images, performed bias field correction, reconstructed cortical surfaces, and registered data to both an undistorted subject native space and the Conte69 standard surface. The functional pipelines removed spatial distortions, corrected motion, registered fMRI data to the structural images, and projected timeseries into the standardized CIFTI grayordinate space (91,282 cortical and subcortical grayordinates). Diffusion preprocessing normalized B0 intensities, corrected susceptibility, eddy current and motion artifacts, applied gradient nonlinearity correction, aligned diffusion data to the structural images, and resampled them to 1.25 mm resolution. Together, these pipelines produced harmonized surface-based multimodal data optimized for cross-subject analyses. QC combined automated metrics with visual inspection at several stages. Automated QC evaluated raw data and intermediate outputs using measures such as framewise displacement, head motion, and changes in global signal, and included checks for protocol conformance and correct frame counts. Trained technicians also visually inspected structural, functional, and diffusion images, as well as surface reconstructions and registration outputs, to identify artifacts or processing failures.

To accommodate the lower tolerance of younger participants for long scanning durations, the HCP-D protocol was condensed into two ~1-h sessions[52]. All acquisitions were performed on standard Siemens 3T Prisma scanners equipped with 80 mT/m gradients and 32-channel head coils. Structural imaging (T1w and T2w) was acquired at 0.8 mm isotropic resolution. Notably, the HCP-D protocol utilized multi-echo MPRAGE and embedded volumetric navigators for real-time prospective motion correction and selective reacquisition of motion-corrupted $k$-space lines. Functional scans (resting-state and task) maintained 2.0 mm isotropic resolution with a TR of 800 ms, utilizing AP/PA phase encoding to optimize performance on the Prisma platform. Diffusion imaging was shortened to ~21 min using a two-shell protocol ($b = 1\,500$ and $3\,000$ s/mm²) at 1.5 mm resolution, with an increased multiband factor (MB = 4) and a shortened TR (3.23 s) to maintain high $q$-space sampling efficiency. Data were processed using the HCP Minimal Preprocessing Pipelines, ensuring harmonized CIFTI grayordinate outputs (91,282 grayordinates) consistent with the HCP-YA workflow. QA was enhanced by real-time motion monitoring (FIRMM) during BOLD acquisitions, complemented by automated metrics (e.g., framewise displacement) and rigorous visual inspection of surface reconstructions and registrations.

Following these preprocessing steps, we constructed individual-level FC and SC matrices using the Surface-Based Connectivity Integration pipeline, a surface-based, atlas-free approach[53]. This approach uses the white surface (the interface between white and gray matter) to construct connectivity matrices without being constrained by pre-defined brain parcellations.

In this framework, FC was obtained from resting-state fMRI by mapping blood oxygen level-dependent time series continuously onto each vertex of the reconstructed white-gray matter boundary in an atlas-free framework[53]. Pairwise Pearson correlations between all vertex-wise time courses were computed and Fisher-z transformed to yield a dense, vertex-wise FC map that captured fine-grained synchrony across the cortical surface without reliance on prespecified parcellations. By treating FC as a continuous field on the white surface, this approach preserved the full topographic detail of functional network organization and facilitated direct, high-resolution comparisons with SC.

Building upon this continuous FC representation, SC was derived from diffusion MRI by estimating fiber orientation distributions via

multi-tissue constrained spherical deconvolution and performing surface-enhanced tractography with seeds placed directly on the white-matter surface mesh[53]. Each streamline's endpoints were projected onto cortical vertices to construct a continuous density function of white matter connections, which was then smoothed and normalized to form a probabilistic, vertex-wise SC matrix perfectly aligned with the FC topology. This continuous SC map provided an anatomical scaffold that could be integrated seamlessly with the FC field for SF coupling analyses.

## Data harmonization

Throughout our analyses, cross-cohort comparisons of SFGC between children (ABCD) and young adults (HCP-YA) were central for revealing developmental differences in gradient coupling. Because such comparisons can be confounded by differences in study protocols, scanners, and sites, we implemented a comprehensive harmonization procedure to align SFGC measures across cohorts. This approach was designed to minimize pipeline- and site-related biases, so that the differences we reported more specifically reflected inherent developmental differences rather than technical artifacts.

It is important to note that directly aligning ABCD and HCP-YA could obscure true cross-cohort differences in the SFGC, because such a procedure cannot distinguish between differences arising from processing pipelines and genuine SFGC differences between the two cohorts. Therefore, we included the HCP-D dataset[54], a companion study to HCP-YA that extends the HCP framework to participants aged 5–21 years. HCP-D uses imaging protocols and processing pipelines closely aligned with those of HCP-YA and samples an age range that overlaps with ABCD. For these reasons, HCP-D served as a bridging dataset between ABCD and HCP-YA. Our procedure thus involved (i) a harmonization step within each cohort to correct for scanner and site effects, and (ii) an alignment step between ABCD and HCP-YA through HCP-D to correct for study and pipeline effects.

First, we applied strict motion quality control by excluding participants with large mean framewise displacement (mean_FD > 0.3 mm), yielding final samples of 5343 ABCD, 93 HCP-D, and 875 HCP-YA participants. Second, we harmonized data within cohorts using ComBat[55] as implemented in the R package sva[56]. For ABCD, we modeled scanner and site as batch variables; for HCP-D, which was collected on a single scanner, we modeled site only. Consistent with standard ComBat practice, age and sex were included as biological covariates so that their effects were preserved while scanner and site effects were removed. HCP-YA was collected on a single scanner at a single site and therefore did not require scanner/site harmonization. Lastly, we corrected for the remaining effects caused by study pipeline differences between ABCD and HCP-D. To achieve this goal, we performed an additional ComBat harmonization treating study (ABCD or HCP-D) as batch effect and again included age and sex as covariates. This would enable us to estimate the mean study effect difference between ABCD and HCP-D, denoted as $\Delta_{\text{ABCD-HCP}}$. Then we added the mean study effect $\Delta_{\text{ABCD-HCP}}$ to each HCP-YA subject, thereby placing ABCD and HCP-YA in a comparable SFGC space while preserving inherent developmental differences rather than pipeline artifacts.

For consistency, we applied this harmonization procedure before using gradients, macroscale SFGC, or subnetwork SFGC in all downstream analyses reported in this paper. As shown in Supplementary Fig. 5, this strategy substantially improved harmonization quality, particularly by mitigating pronounced motion- and scanner-related effects in the ABCD cohort.

## Functional and structural gradient construction

Cortical gradients were extracted to capture macroscale organizational principles of structural and functional connectivity using the BrainSpace toolbox[57]. Following previous gradient-based imaging studies[27,36], we applied proportional thresholding at the subject level:

for each individual-level SC and FC matrix, we ranked all non-diagonal connection weights and retained the strongest 10% of edges, setting the remaining 90% to zero. This procedure controls network sparsity and reduces the influence of weak or noisy edges while maintaining comparability across participants and modalities. We then transformed these sparsified matrices into cosine similarity affinity matrices for gradient estimation. Gradients were then derived via diffusion mapping, a nonlinear dimensionality reduction technique known for its robustness to noise. In short, diffusion map embedding constructs a Markov operator $P^{(\alpha)} = (\Gamma^{(\alpha)})^{-1} W^{(\alpha)}$, where $\Gamma^{(\alpha)}$ is a diagonal matrix with $\Gamma_{i,i}^{(\alpha)} = \sum_j W_{i,j}^{(\alpha)}$, $W^{(\alpha)} = \Gamma^{-1/\alpha} A \, \Gamma^{-1/\alpha}$ is the affinity matrix $A$ reweighted by the anisotropic diffusion parameter $\alpha$, and $\Gamma$ is the degree matrix of $A$. One then solves $P_\alpha \phi_\ell = \lambda_\ell \phi_\ell$, and defines the $m$-dimensional embedding at diffusion time $t$ (the number of virtual steps in the random walk) as

$$\mathcal{G}_{\text{DM}}(i) = (\lambda_1^t \phi_1(i), \lambda_2^t \phi_2(i), \ldots, \lambda_D^t \phi_D(i)), \tag{1}$$

omitting the trivial $\lambda_0 = 1$ eigenvector. Raising each eigenvalue to the $t$th power downweights modes that decay quickly. This yields gradients that capture both local and global connectivity structure with built-in noise robustness. In our implementation, we set $\alpha = 0.5$ to balance the influence of sampling density, ensuring that neither dense nor sparse regions dominate the embedding. We used $t = 0$ to preserve the full spectrum of connectivity detail.

To enable group-wise comparisons, we aligned individual gradients to a cohort-level template, thereby minimizing inter-subject geometric variability. Following this pipeline, we retained the first 200 gradients per modality, which together explained over 99% of the variance in connectivity structure. For subject $i$, the resulting functional gradients $G_i^f$ and structural gradients $G_i^s$ were matrices of dimension $V \times D$, where $V$ denotes the number of cortical vertices and $D = 20$ is the number of retained diffusion components, selected according to the highest eigenvalues in descending order. The resulting principal gradients in both the ABCD and HCP cohorts closely resembled previously reported macroscale gradient topographies[14,36,57,58].

In the cohort-level maps (Fig. 2a, g), the first two functional gradients accounted for 39% of variance in adults and 43% in children. In contrast, the first two structural gradients explained 49% of variance in adults but only 22% in children. While most information was captured by the first few components, variance explained declined gradually across higher-order gradients, dropping by less than 10% per gradient, and less than 5% beyond the fifth. To balance interpretability with information retention, we retained gradients that collectively explained at least 70% of the total variance. Specifically, in adults, the top 20 gradients accounted on average for 74% of structural and 92% of functional connectivity variance. In children, the corresponding figures were 83% for structural and 70% for functional connectivity. Note that the specific topographical ordering of sub-dominant gradients (e.g., FCG2 in adults) can be sensitive to the spatial representation of the connectome and the close proximity of eigenvalues, which is consistent with perturbation theory for eigenspace stability.

For gradient analyses, we performed two sets of statistical tests. First, to assess cross-cohort spatial correspondence of cohort-level gradients, we conducted spin-rotation permutation tests (10,000 permutations) separately for FCGs and SCGs. For each gradient pair (e.g., FCG$i$ in adults and FCG$j$ in children), we evaluated the spatial correspondence between gradients from the two cohorts and assessed its significance against a spatially constrained null distribution generated by spin permutations, with p-values corrected for multiple comparisons using FDR. Second, to assess hemispheric asymmetry in structural gradients, we performed two-sided paired $t$-tests for each SCG within each cohort.

## Structural-functional gradient coupling

To quantify the alignment between SC and FC, we introduced SFGC, based on the gradients derived from each modality. We evaluated this coupling at two levels: (i) macroscale, capturing whole-brain gradient alignment, and (ii) subnetwork, assessing regional specificity (see Fig. 1 for illustration).

At the macroscale level, for each subject $i$, we computed the SFGC $\Psi_{idd'}$ as the cosine similarity between the $d$th column of the functional gradient matrix $G_i^f$ and the $d'$ th column of the structural gradient matrix $G_i^s$, defined as:

$$\Psi_{idd'} = \frac{\left\langle G_{i,\cdot d}^f, G_{i,\cdot d'}^s \right\rangle}{\left\| G_{i,\cdot d}^f \right\| \left\| G_{i,\cdot d'}^s \right\|}, \tag{2}$$

where $G_{i,\cdot d}^f \in \mathbb{R}^V$ is the $d$th functional gradient for all $V$ vertices; $G_{i,\cdot d'}^s \in \mathbb{R}^V$ is the $d'$ th structural gradient for all $V$ vertices; $V$ is the number of vertices in the whole brain; $\langle \cdot, \cdot \rangle$ denotes inner product of two vectors; and $\| \cdot \|$ denotes the standard Euclidean norm of vectors. The resulting covariate vector $\boldsymbol{\Psi}_i = (\Psi_{i11}, \ldots, \Psi_{iDD})$ is of length $D^2$, representing individual-specific SF gradient alignment. For example, selecting the top $D = 20$ gradients from each modality yields a coupling vector of length 400. Furthermore, we evaluated differences in absolute SFGC between cohorts for each gradient pair using two-sample Welch's t-tests, with p-values adjusted using the FDR approach.

To capture regional variation, we also constructed subnetwork-level SFGC, parcellating coupling signals according to two atlases: the Yeo-7 network functional parcellation[19] and the D-K structural atlas[26]. Specifically, the gradients $G_i^f$ and $G_i^s$ were first partitioned into $J$ subvectors corresponding to the $J$ predefined regions of interest (ROIs), denoted as $G_{i,jd}^s \in \mathbb{R}^{V_j}$ and $G_{i,jd'}^f \in \mathbb{R}^{V_j}, 1 \le j \le J, 1 \le d, d' \le D$, respectively. Here, $V_j$ is the number of vertices within the $j$th ROI. Then the subnetwork-level SFGC $\Psi_{ijdd'}$ was calculated as the cosine similarity between each pair of functional and structural gradients within each of the $J$ ROIs, defined as

$$\Psi_{ijkk'} = \frac{\left\langle G_{i,jd}^f, G_{i,jd'}^s \right\rangle}{\left\| G_{i,jd}^f \right\| \left\| G_{i,jd'}^s \right\|}. \tag{3}$$

This yielded a subnetwork-level SFGC of dimension $d_{sf} = J \times D^2$. For example, using the Yeo-7 network with $J = 14$ and $D = 20$ results in a $d_{sf} = 5\,600$-dimensional vector per subject. It is important to note that the association studies for different subnetworks were jointly performed. These subnetwork couplings captured localized SF alignment within canonical brain systems.

## Associations between SFGC and behavioral outcomes

To examine associations between SFGC and behavioral outcomes, KRR and MLP models were applied independently to each SFGC-outcome pair, using either the macroscale or subnetwork-level SFGC vectors as input features.

In the KRR model[20], for each subject $i$, we denoted the SFGC vector as $\mathbf{x}_i \in \mathbb{R}^{d_{sf}}$, and the outcome variable as $y_i$. Note that $\mathbf{x}_i$ was a vector of macroscale SFGC with dimension 400 or Yeo-7 subnetwork SFGC with dimension 5,600, and was standardized element-wisely for model stability. The model minimized the regularized squared loss: $\frac{1}{n} \| \mathbf{y} - \boldsymbol{\psi}\mathbf{w} \|^2 + \lambda \| \mathbf{w} \|^2$, where $n$ was the number of subjects; $\boldsymbol{\psi} \in \mathbb{R}^{n \times n}$ was a matrix with linear kernel basis of the form $\psi(\mathbf{x}_i, \mathbf{x}_j) = \mathbf{x}_i^T \mathbf{x}_j, 1 \le i, j \le n$; $\mathbf{w} \in \mathbb{R}^n$ were the coefficients to be estimated; $\lambda$ was a regularization parameter selected via cross-validation from the set $\{0.01, 0.1, 0.5, 1, 5, 10, 50, 10^2, 10^3, 10^4, 10^5\}$. Note that a higher $\lambda$ would inflate the penalty term $\lambda\|\mathbf{w}\|$ and lead to a sparser model.

For MLP, we first introduced the concept of neural networks. A neural network is a computational model inspired by the human brain's structure, designed to recognize patterns, learn from data, and make predictions. It consists of layers of interconnected nodes or "neurons," including an input layer that receives data, hidden layers that process it, and an output layer that delivers the final results. Each connection has associated weights and biases, which the network adjusts during training to optimize performance. Using activation functions, neural networks introduce non-linearity, enabling them to model complex relationships. The learning process involves forward propagation, where data flows through the network to produce an output, and backpropagation, where errors are used to update weights via optimization algorithms like gradient descent. In our analysis, we implemented MLPs with three fully connected layers for the prediction tasks. The input layer consisted of the macroscale or subnetwork-level SFGC. The hidden dimension was set to 128, with a dropout rate of $p = 0.5$ to mitigate overfitting, and leaky ReLU activation with a negative slope of 0.2 was applied to the first two layers. For sex prediction, an additional Sigmoid function was included to map the outputs to the range [0, 1]. The number of training epochs was selected via cross-validation, with a maximum of 50. In the main text, we focused on the KRR results, and presented the MLP results in Supplementary Fig. 2 as a robustness check, which shows strong consistency.

## Model performance and feature importance

We evaluated the model performance by comparing each observed behavioral outcome and the predicted behavioral outcome using SFGC. Specifically, for continuous outcomes of interest (e.g., CogTotal), we computed the out-of-sample Pearson correlation coefficient ($r$) between observed and SFGC-predicted outcome variable for each of 100 splitting replicates. For binary outcomes (e.g., sex), we computed the area under the receiver operating characteristic curve (AUC) using observed labels and SFGC-predicted label probabilities for each of the 100 splitting replicates. The resulting 100 correlation coefficients or AUCs for each coupling-outcome correspondence were averaged or used for two-sample paired t-tests and Welch's t-tests in SFGC-outcome association studies in the Results section. This procedure of utilizing 100 splitting replicates and computing out-of-sample correlation coefficients and AUCs reinforced the robustness of the discovered association between the SFGC and outcomes of interest.

To assess feature relevance, we computed attribution scores for each model. For KRR, raw feature importance was computed as the primal coefficients, i.e., the correlation coefficients with respect to SFGC, denoted as: $FI_0 = \mathbf{x}^T \hat{\mathbf{w}} \in \mathbb{R}^{d_{sf}}$, where $\hat{\mathbf{w}}$ was the learned weights of the kernel functions $\boldsymbol{\psi}$; $\mathbf{x} = (\mathbf{x}_1, \cdots \mathbf{x}_n) \in \mathbb{R}^{n \times d_{sf}}$ was the concatenated SFGC from all subjects; $d_{sf}$ was the dimension of SFGC features, $d_{sf} = 400$ when macroscale SFGCs were used, $d_{sf} = 5600$ when subnetwork-level SFGCs under Yeo-7 network were used. For MLP, we used a saliency map[59], defined as the average gradient of the output with respect to the input features: $FI_0 = \frac{1}{n}\sum_{i=1}^n \nabla_{\mathbf{x}_i} f(\mathbf{x}_i) \in \mathbb{R}^{d_{sf}}$, where $\nabla_{\mathbf{x}_i} f(\mathbf{x}_i)$ indicated the partial derivative vector of the outcome of interest $f(\mathbf{x}_i)$ with respect to SFGC $\mathbf{x}_i$. Then feature importance (FI) was defined as min-max normalized raw importance:

$$FI = \frac{|FI_0| - \min(|FI_0|)}{\max(|FI_0|) - \min(|FI_0|)} \in [0,1]^{d_{sf}}. \tag{4}$$

Higher normalized values indicated stronger predictive contributions of the corresponding features. It is important to note that we concatenated the subnetwork SFGC from all subnetworks, i.e., we use features of a dimension $5\,600 = 400$ SFGCs $\times 14$ subnetworks in the Yeo-7 atlas. The feature importance from different subnetworks was extracted from a joint model fitting and was thus comparable.

## Heritability estimation using the ACE model

We estimated the heritability of functional and structural gradients, as well as SFGC, using the classical ACE model based on twin and sibling data from the HCP and ABCD cohorts. The phenotype for each subject was modeled as:

$$\mathbf{y} = \mathbf{X}\boldsymbol{\beta} + \mathbf{g} + \mathbf{c} + \mathbf{e}, \tag{5}$$

where $\mathbf{y} \in \mathbb{R}^N$ was the phenotype vector for $N$ individuals, $\mathbf{X} \in \mathbb{R}^{N \times Q}$ denoted the design matrix of fixed covariates (including age and sex), and $\boldsymbol{\beta} \in \mathbb{R}^Q$ represented the corresponding fixed effects. The random effects included the additive genetic effect $\mathbf{g} \sim \mathcal{N}(\mathbf{0}, \sigma_A^2 \mathbf{K})$, the shared environmental effect $\mathbf{c} \sim \mathcal{N}(\mathbf{0}, \sigma_C^2 \mathbf{\Lambda})$, and the unique environmental effect $\mathbf{e} \sim \mathcal{N}(\mathbf{0}, \sigma_E^2 \mathbf{I})$. Here, $\mathbf{K}$ was the genetic relatedness matrix and $\mathbf{\Lambda}$ represented the shared environment matrix, both constructed from zygosity and parental identifiers. Heritability was defined as the proportion of phenotypic variance attributable to additive genetic effects:

$$h^2 = \frac{\sigma_A^2}{\sigma_A^2 + \sigma_C^2 + \sigma_E^2}. \tag{6}$$

We used maximum likelihood estimation to fit the ACE model, where the variance components $\sigma_A^2, \sigma_C^2$, and $\sigma_E^2$ were estimated from the residuals of the fixed-effect model after regressing out the effects of age and sex. The ACE model was applied vertex-wise for functional and structural gradients, and region-wise for SFGC at both subnetwork (Yeo-7) and regional (D-K atlas) levels, given that the D-K atlas enabled a more precise characterization of the spatial distribution of heritability and allowed for direct alignment with transcriptomic data. For the young adult cohort, we used the full sample, which includes MZ and DZ twins, full non-twin siblings, and extended family structures sufficient for estimating genetic and environmental variance components. In contrast, for the children cohort, we restricted the heritability analyses to a genetically informative subset consisting of siblings only ($n = 975$), rather than using the full sample of over 5000 participants. This restriction was motivated by the extremely sparse genetic and shared environment matrices derived from the full sample, which would hinder the identifiability and stable estimation of variance components. By focusing on siblings with known zygosity, we ensured that the model assumptions were satisfied and that heritability could be robustly estimated.

We evaluated the statistical power to detect nonzero narrow-sense heritability using simulations tailored to each cohort's observed covariate matrix $\mathbf{X}$ and relatedness structures $\mathbf{K}$ and $\mathbf{\Lambda}$. Phenotypes were simulated under the model $\mathbf{y} \sim \mathcal{N}(\mathbf{X}\boldsymbol{\beta}, \sigma_A^2 \mathbf{K} + \sigma_C^2 \mathbf{\Lambda} + \sigma_E^2 \mathbf{I})$ with total variance normalized to one, for a grid of assumed true heritabilities ($h^2 = 0.01 - 0.80$) and varying proportions of shared environmental variance $\left(\frac{\sigma_C^2}{\sigma_C^2 + \sigma_E^2} = 0, 0.3, 0.5, 0.8\right)$. For each parameter combination, we fitted both the full ACE model and a reduced model with $\sigma_A^2 = 0$ on 5000 simulated datasets, and calculated likelihood-ratio statistics using the standard 50:50 mixture of $\chi_1^2$ distributions appropriate for boundary-limited variance components. Power was defined as the fraction of simulations yielding $p < 0.05$. Power curves summarizing these results were provided in Supplementary Fig. 6.

To enable direct statistical comparison across modalities, unimodal FC and SC gradients were summarized at the same subnetwork resolution used for SFGC (Yeo-7 and D-K). For each participant, we extracted subnetwork-level gradient values by averaging vertex-wise gradient scores within each subnetwork. SFGC values were naturally defined at this same resolution. We computed the heritability of each subnetwork-level phenotype and compared SFGC to its matched FC or SC gradient using two-sided paired t-tests across subnetworks. Because FC, SC, and SFGC were all summarized to the same subnetwork resolution, each subnetwork provided a matched phenotype across modalities, enabling valid paired-sample comparisons. Two comparison schemes were implemented: (i) a global metric aggregating the first 20 unimodal gradients and (ii) principal gradient-specific comparisons (the first three gradients). All paired t-test p-values were corrected for multiple comparisons using the Bonferroni procedure (reported in Supplementary Table 1).

## Imaging transcriptomics of gradient coupling heritability

To further investigate the biological mechanisms underlying the observed heritability patterns in SFGC, we performed imaging transcriptomic analyses that integrated regional heritability estimates with gene expression profiles. Analyses focused on the SFGC between the first FCG and SCG (FCG1:SCG1), which represented the principal axes of cortical organization and relatively high heritability. For the young adult cohort, we utilized adult gene expressions from AHBA[60], which contained 15,633 genes across 66 cortical ROIs (two ROIs with missing gene data were excluded), and adult cortical single-cell transcriptomics data[61]. Following the established imaging transcriptomics workflow[62], we extracted region-wise gene expression profiles from the AHBA using the *abagen* toolbox[63], mapped to the D-K atlas[26]. Gene expression values were normalized within each region to a total count of 10,000. For the children cohort, we used adolescent gene expression data from the BrainSpan atlas[64] (21,315 normalized genes across 11 ROIs) and corresponding adolescent single-cell transcriptomic data[65]. We computed Spearman's rank correlations between the heritability values and gene expressions across ROIs, and ranked genes in descending order based on their correlation with the phenotype-specific heritability pattern.

To account for spatial autocorrelation in both the gene-level correlations and subsequent enrichment analyses, we generated 10,000 spin-rotation permutations of the regional heritability map. Observed statistics were compared against the corresponding null distributions derived from these permutations to obtain spatially adjusted p-values (Supplementary Table 4). All gene-heritability correlations and GSEA enrichment statistics were evaluated relative to the empirical null from these spin permutations, yielding spatially adjusted significance values.

To assess cell-type specificity, we performed GSEA using the *GSEApy* library[66]. Cell-type-specific gene sets were derived from scRNA-seq data, where the top 200 differentially expressed genes for each cell type were identified using t-tests implemented via the *SCANPY* pipeline[67]. GSEA was applied to calculate enrichment scores based on the ranked gene list for each phenotype (e.g., heritability of the first functional gradient or SFGC).

## Analysis robustness and stability

For robustness evaluation, we generated 100 replicates of the data splits; within each replicate, model performance was assessed using 10-fold cross-validation. Family structure was accounted for by assigning participants from the same family to the same cross-validation fold. Since we obtained 100 realizations of SFGC-outcome associations and 100 normalized feature importance scores for each SFGC-outcome pair, these replications enabled us to perform two-sample paired t-tests for comparing SFGC-outcome association difference between macroscale SFGC and subnetwork SFGC; and Welch's t-test for comparing SFGC-behavior association difference between cohorts and between sex in downstream analyses. This procedure ensured the robustness of the SFGC-outcome correspondence and mitigated potential issues of overfitting and sampling bias.

The two predictive modeling approaches, KRR and MLP, demonstrated strong concordance. We presented the association results based on MLP in Supplementary Fig. 2, which were highly comparable association patterns to Fig. 3a produced by KRR. In addition, the overall Pearson's r between KRR and MLP in the SFGC-

behavior association was 0.75 for macroscale SFGC, and 0.80 for subnetwork SFGC, highlighting the robustness of the associations we identified. Furthermore, KRR and MLP exhibited similar patterns of feature importance, where the most important SFGC features identified by the two models shared similar patterns for both macroscale SFGC (Supplementary Fig. 3a) and subnetwork SFGC (Supplementary Fig. 3b). The averaged feature importance values, computed across both methods, were reported in the Results section.

### Reporting summary

Further information on research design is available in the Nature Portfolio Reporting Summary linked to this article.

## Data availability

Neuroimaging and behavioral data from the ABCD Study can be obtained via the NIH Data Archive (https://nda.nih.gov/abcd) with approval from the ABCD consortium. Neuroimaging data and most behavioral measures from the HCP-YA are publicly available at https://db.humanconnectome.org; access to restricted data is subject to approval. Data from the HCP-D study are available through the NIH Data Archive (https://nda.nih.gov) and require approval for access. The raw data are protected and are not available due to data privacy laws and the terms of the original ethical approvals. Source data are provided with this paper. The data are supplied in multiple formats, including Python pickle (.pkl) files, which can be read using standard Python packages such as `pickle` or `pandas`. Source data are provided with this paper.

## Code availability

The data preprocessing software FreeSurfer v6.0 is available at https://surfer.nmr.mgh.harvard.edu/. The Surface-Based Connectivity Integration pipeline can be accessed at https://github.com/sbci-brain/SBCI_Pipeline, and the BrainSpace toolbox is available at https://github.com/MICA-MNI/BrainSpace/tree/master. Python (2.7 and 3.12) and R v4.4 were used for data processing and analysis. Code used in this study is publicly available at https://github.com/Zhao-team/SF-Gradient-Coupling.git and stored at https://doi.org/10.5281/zenodo.18912522[68].

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

## Acknowledgements

S.G., Z.G., S.D., G.W., and Y.Z. were partially supported by National Institutes of Health (NIH) grants R01AG068191, RF1AG081413 and R01EB034720 to Y.Z. We express our sincere gratitude to the participants and researchers of the ABCD Study and the HCP, and gratefully acknowledge the use of data from both consortia in this research. ABCD data were obtained from the ABCD study, held in the National Institute of Mental Health Data Archive. This is a multisite, longitudinal study designed to recruit more than 10,000 children age 9–10 years and follow them over 10 years into early adulthood. The ABCD study is supported by the NIH and additional federal partners under award numbers U01DA041022, U01DA041028, U01DA041048, U01DA041089, U01DA041106, U01DA041117, U01DA041120, U01DA041134, U01DA041148, U01DA041156, U01DA041174, U24DA041123, U24DA041147, U01DA041093 and U01DA041025. A full list of supporters is available at https://abcdstudy.org/federal-partners.html. A listing of participating sites and a complete listing of the study investigators can be found at https://abcdstudy.org/scientists/workgroups/. ABCD consortium investigators designed and implemented the study and/or provided data but did not necessarily participate in analysis or writing of this report. All procedures in the ABCD study were approved by the institutional review boards at ABCD collection sites (approval numbers 201708123 and 160091). HCP-YA and HCP-D data were provided by the HCP, WU-Minn Consortium (principal investigators

D. Van Essen and K. Ugurbil; 1U54MH091657) funded by the 16 NIH institutes and centers that support the NIH Blueprint for Neuroscience Research and the McDonnell Center for Systems Neuroscience at Washington University. All experimental procedures in the HCP were approved by the institutional review boards at Washington University (approval number 201204036).

## Author contributions

These authors contributed equally: Simiao Gao, Zhiling Gu, Shengxian Ding. Y.Z., S.G., Z.G., and S.D. conceptualized and designed the study. Z.Z. and H.Z. collected and processed the data. S.G., Z.G., S.D., and G.W. analyzed the data. S.G., Z.G., and S.D. wrote the initial draft of the manuscript, and all authors edited and reviewed the final manuscript.

## Competing interests

The authors declare no competing interests.
