## [Transparent Peer Review file · Nature Communications]

Brain Functional-Structural Gradient Coupling Reflects Development, Behavior and Genetic Influences

Corresponding Author: Dr Yize Zhao

Version 0:

Reviewer comments:

Reviewer #1

(Remarks to the Author)

The study by Gao et al., examines how the brain structural-functional (S-F) gradient differs between two age-stratified cohorts (i.e., ABCD and HCP-YA cohorts). The results showed cohort and sex-specific differences in SF coupling, associations between SF gradient coupling and cognitive and mental health metrics, spatial overlap between SF gradient coupling and gene expression profiles from AHBA indicating enrichment for genes expressed in deep-layer excitatory neurons, and provide regional heritability estimates for the gradient coupling metrics.

The question of how structural functional gradient coupling evolves during development and relates to behavioral, genetic and molecular features is important and somewhat understudied, to my knowledge. As such, strengths of the study are its comprehensiveness and utilization of multimodal MRI across two well-characterized, large datasets of different ages. The paper is, in general, clearly written. However, there are some major substantive concerns, as well as several minor concerns, regarding the literature review, claims and interpretation of the findings and some of the analyses, which diminish enthusiasm for the manuscript in its current form:

1. A more comprehensive description of the previous literature should be included in the introduction and discussion. Despite methodological differences, the S-F coupling gradient has been characterized and examined across several papers previously, which also include replication datasets. E.g., Development of S-F coupling gradient using cross-sectional and longitudinal data, as well as examining associations to cognitive performance (Baum et al., 2019, PNAS, doi: 10.1073/pnas.1912034117); S-F coupling gradient in young adults, with HCP as a replication dataset (Vazquez-Rodriguez et al., 2019, PNAS, doi: 10.1073/pnas.1903403116); age-related differences S-F coupling assessed across the lifespan (Esfahlani et al., 2022, Nat Com, doi: 10.1038/s41467-022-29770-y); characterization of the S-F coupling gradient with replication dataset (Popp et al., 2024, NeuroImage, <https://doi.org/10.1016/j.neuroimage.2024.120563>); S-F coupling, age-related differences and spatial overlap with gene expression (Feng et al., 2024, eLife, <https://doi.org/10.7554/eLife.93325.2>); discussed in a review article (Suarez et al., 2020, Trends in Cognitive Sciences, doi: 10.1016/j.tics.2020.01.008). While these articles are cited, the characterization in the introduction of the novelty of this study should be revised. Furthermore, the SNP based heritability of the S-F coupling also has been examined previously by some of the same authors using the same datasets as the current paper (Dai et al., 2024, Imaging Neuroscience, https://doi.org/10.1162/imag_a_00346). Some of these papers also examined S-F coupling changes in sensory and associations areas across childhood and adolescence (i.e., Feng) and across the lifespan (i.e., Esfahlani), which do not fit with the authors claims in the introduction (i.e., “no study has yet tested how SF gradient coupling evolves from late childhood through early adulthood”). Thus, it would be of importance to properly put these findings in the context of previous studies.

2. Although I find most of the analyses to be appropriate, some of the claims seem broader than what the results justify. As acknowledged by the authors, any differences between the two cohorts can also be attributed to other characteristics than age, such as sample characteristics (e.g., socioeconomic status, ethnicity, scanner effects, image quality). Thus, the paper would benefit from rephrasing words like “trajectories” and “evolved”, which gives the impression of a longitudinal analysis and causality.

3. As a related point, the results are often presented in a ‘big picture’ summary style (e.g., p5, ‘Their second functional gradient began to resemble the adult principal axis, indicating...’) This statement is easy to understand, but what particular analysis supported this conclusion? Likewise, on p 6 there is a statistic for SF gradient coupling in children, and a lower C-

value for adults, but from what I can tell, there was no direct statistical comparison conducted. There is more quantitative information presented in the figures and the supplements, but even there there is a fair amount of un-evenness in the level of detail provided about what is being depicted (e.g., what conclusions should be drawn from Supp Figures 1 and 3?). More detailed, quantitative information for each of the results and figures would be helpful in evaluating the strength of the claims made.

4. In relation to cohort differences, it would be helpful if the authors addressed the within sample correlations in more depth, as within-cohort variability may influence the brain-behavior relationships within each cohort. For instance, is there more variance in the cognitive and mental health metrics in the ABCD vs HCP-YA? If so, could the authors elaborate on the potential impact of range restrictions on the brain-behavior associations?

5. The authors state that 'the potential of SF gradient coupling as a predictive feature for behavioral traits has not been investigated'; to that end they employ two predictive modeling approaches to examine relationships to mental health symptoms and cognition, but from what I can tell they only utilized cross-sectional ABCD data. Why not utilize the longitudinal data to conduct an actual predictive analysis?

6. The ABCD dataset includes MRI data across >20 sites, and it is unclear to me how the authors addressed potential differences across scanner sites. Examination of how SF coupling may differ across the scanner sites can provide important insight into the potential confounding variables within the ABCD cohort. In addition, younger children also tend to move more in the scanner, which can also influence data quality. Thus, it would be helpful if the authors compared the image quality between the two cohorts (e.g., through Euler number/SNR/tSNR/framewise displacement) to rule out the potential impact of image quality on SF coupling.

7. How did the authors address potential type 1 errors due to spatial autocorrelation between the brain maps and transcriptomic maps? The wording of the transcriptomics analyses reflects a reverse inference logic, i.e. "Transcriptomic analyses further demonstrated that highly heritable coupling patterns are enriched for genes expressed in deep-layer excitatory neurons" "Transcriptomic signatures highlight the cellular basis of SF gradient coupling heritability", "To identify specific molecular contributors", "To explore the cellular basis of heritability variation in SF gradient coupling", "transcriptomic analyses revealed that the heritability of gradient coupling was enriched for excitatory neuronal signatures" this type of phrasing could be perceived as too strong given the data; a more neutral tone might be more appropriate (e.g., emphasizing the spatial overlap with gene expression maps that are enriched for genes that are highly expressed in deep layer neurons). In addition, since the gene expression values derived from AHBA are based on post-mortem data from adults, it is unclear to me how this would be an appropriate analysis to perform on brain maps derived from children. In addition, this section suffers from the same issues noted above regarding broad statements and lack of precision, e.g. "Particularly strong signals were observed for excitatory neurons 3 and 4.." What is considered the threshold for 'particularly strong'? was that determined a priori ?

Minor:

8. The study lacks information about the power analyses for the heritability estimates, which will be important given the sample size differences in the two cohorts.

9. The authors state that they performed two-sample t-tests, but it is unclear if they are assuming equal or unequal variance between the groups.

10. It would be helpful if the authors provide the age-range of the final analytic samples and a demographic table stratified by sex.

11. "Our approaches and results revealed that SF gradient coupling undergoes systematic neurobiological refinement from late childhood to early adulthood with distinct sex-specific trajectories."

This statement could benefit from more careful wording as it gives the impression of causality and longitudinal analyses.

12. 'Finally, we integrated transcriptomic data from the AHBA to reveal that heritable gradient coupling patterns are enriched for genes expressed in deep-layer excitatory neurons'

The authors should clarify that these analyses examine the spatial overlap between brain and transcriptomic maps.

13. 'Rather than characterizing the brain as a set of discrete networks, gradient-based approaches project connectivity patterns into low-dimensional manifolds that reveal continuous axes of organization.'

It is not clear to me what the authors mean by 'low-dimensional manifolds', I suggest rephrasing for clarity.

14. 'Even fewer studies have examined how gradient coupling develops over time or varies across individuals.'

It would be helpful if the authors provided references to these studies. See major comment.

15. 'Although many aspects of brain development unfold gradually across childhood and adolescence, most studies of SF coupling have focused on either mature adult brains or aggregated developmental trajectories.'

Could the authors clarify on what they mean by "aggregated developmental trajectories"? It is also hard to evaluate this claim without references.

16. 'Specifically, we first thresholded the SC and FC matrices to retain the top 10% of connections, then transformed them into cosine similarity affinity matrices.'

Could the authors elaborate on how they define the top 10% connections?

17. 'We further observed that the SF gradient coupling strength between the principal functional gradient and the second structural gradient (FCG1:SCG2) increased from near-zero levels in children to a robust level in adults'

It would be helpful if the authors provide references to the figure or table that support this claim.

18. 'Furthermore, the association between SF gradient coupling and outcomes varied by scale and age cohort: in adults, subnetwork-level SF gradient coupling showed stronger association with psychiatric symptoms than macroscale gradient coupling, whereas in children, macroscale SF gradient coupling yielded exhibited associations.'

It would be helpful if the authors provided a reference to a figure or numbers that supports this claim. It is also unclear to me which statistical test and results that was used to support the claim of a 'stronger association'. Also, it is unclear to me what is meant by "yielded exhibited associations" ?

19. For SF gradient coupling and cognitive performance scores, the authors state that associations get weaker in adults, but I struggle to find the statistical test was performed to support this claim other than a numerically lower correlation. In addition, the wording gives the impression that differences in correlations are due to age (e.g., "became weaker in adults") – if there is a significant difference, a more accurate description would be "lower in the adult cohort".

20. 'After rigorous quality control and preprocessing of the neuroimaging data, and exclusion of participants with incomplete covariate information, the final analytic sample consisted of 7,025 subjects (mean age = 9.9 years; 48.8% male).'

'Following quality control and data aggregation, our final analytic sample comprised 913 participants (mean age = 28.7 years; 46.55% male) from 414 families, including 111 MZ twin pairs, 59 DZ twin pairs, 511 full sibling pairs, and 95 singletons.'

It would be helpful if the authors could elaborate on the quality control procedures performed on each cohort.

21. Will the authors provide the regional values for the SF coupling gradients? This could be further used by other researchers to examine the spatial overlap with SF coupling with other cortical maps.

22. The GitHub link doesn't work. It would be helpful if the authors could open the link for review.

23. p. 3- 'when brain organizations are more stable and specialized"- change to 'when brain organization IS more stable and specialized'

Reviewer #2

(Remarks to the Author)

This paper measures the similarity of functional and structural gradients (SF gradient coupling) in a cohort of children (9-10 years old, ABCD dataset) vs adults (22-35 years old, HCP dataset). The authors assess cohort-level differences in each gradient modality and their coupling across cohorts, with coupling evaluated across all regions ("macroscale") and stratified within specific subnetworks. They show different SF coupling patterns between children and adults spanning several subnetworks. A series of within cohort analyses link SF coupling patterns to mental health and cognitive measures, identifying different magnitude and patterns of associations across cohorts and additionally uncovering sex-specific effects. More specifically, SF coupling and cognitive measure associations were generally stronger in children, while mental health relationships were stronger in adults. Heritability analyses across modality-specific and SF coupling gradients captured overall similar patterns in both groups, with slightly higher heritability estimates in children, particularly for the SF coupling assessment. Transcriptomic decoding of SF coupling heritability estimates showed associations with the expression of excitatory neurons subtypes.

The paper includes an impressive breadth of analyses. However, I have several major concerns, notably regarding the motivation of the study, the way developmental comparisons were implemented, and note that several interpretations and conclusions are not supported by adequate statistical tests.

1 - The introduction lacks a clear rationale for why gradients and SF gradient coupling are used and important for neurodevelopmental inquiry. Currently, the paper is framed as if the primary motivation for looking at SF gradient coupling is

that it has not been done before, and therefore behavioural and genetic associations of SF coupling are unknown. What insights can SF gradient coupling bring beyond previously used measures of FC-SC coupling? What would diverging results from connectome-level measures indicate? Without this justification, it is difficult to contextualize the importance or new insights brought by the multitude of analyses performed here.

2 - The authors perform group-level comparisons of children (9-10) vs young adults (22-35) cohorts. First, as data are not harmonized across sites with methods such as Combat, it is unclear if reported effects are attributable to differences in age or general site/scanner differences. Second, and as the authors know, both white matter and functional networks change throughout the lifespan, and derivative metrics from these networks peak at different ages for different subnetworks. Childhood is a developmental window of rapid change, and even in the adult time window investigated here, FC and SC are not static (Yeatman et al., 2014, Nat Comms; Fjell et al., 2017, HBM; Betzel et al., 2014, NIMG). Most developmental studies model age effects continuously for this reason. As a result, I am not convinced that a cohort-level comparison performed on unharmonized data is appropriate.

3 - Several analytical choices are not adequately justified. For example, some analyses focus only on G1 and G2, the coupling of FCG1 and SCG1 or FCG1 and SCG2, and the heritability analyses look at 20 distinct gradients. It is hard to make sense of these analytical variations and interpret findings coherently. For example, would we expect different downstream findings from measuring SF gradient coupling using FCG1:SCG1 vs FCG1:SCG2?

4 - Many claims in the paper are not statistically supported. For example, differences between young and adult cohorts on FC and SC gradients are not quantified, only described qualitatively. As a result, the effect of modality-specific differences on SF gradient coupling differences cannot be assessed (i.e. are the effects driven by one modality over the other). On pages 6-8, the authors claim "Together, these findings established macroscale and subnetwork-level SF gradient coupling as two equally informative metrics for quantifying correspondence between brain structural and functional organization." Yet, no quantitative comparison of the two approaches is performed – how can they be qualified as equally informative? In the heritability analyses, the authors report "These findings suggested that SF gradient coupling captured highly heritable multimodal axes of brain organization that were not fully explained by either modality alone." They support these claims by simply reporting heritability metrics FC, SC and SF coupling gradients, yet there are no models implemented to assess unique variance explained or direct statistical comparisons of these heritability metrics.

5 - HCP and ABCD gradients were aligned to a "cohort-level" template. By aligning each cohort to distinct templates, does this procedure not run the risk of spuriously inflating group effects?

6 - The use of causally-ambiguous language in describing transcriptomic findings is unwarranted for a correlation approach such as the one applied here.

7 - The Discussion is overall very short and provides little to no contextualization of results with the literature, focusing on a summary of the findings.

8 - The link provided to reproduce the original analyses in this paper is not functional (<https://github.com/Naomi-Ding/SF-Gradient-Coupling> – 404 not found). It is thus not possible to evaluate or review the code used to produce these findings.

9 - As a minor point, the figures are generally quite dense, making the text and smaller figure panels difficult to parse. Also, the magnitude and topography of group differences are difficult to evaluate when adult and child effects are not presented on the same scale (e.g., Fig 1d-j; f-l, but the point holds throughout the manuscript).

Reviewer #3

(Remarks to the Author)

This study introduces and characterizes a novel neuroimaging metric, structural-functional (SF) gradient coupling, which quantifies the alignment between structural and functional gradients both globally and within distinct brain networks. The authors examine how this coupling varies across developmental stages, differs by sex, relates to behavior and mental health, and reflects genetic and transcriptomic influences.

Given that the gradients are derived separately for functional and structural modalities, and thereby their rank order does not necessarily correspond across modalities, the authors opted to explore all possible gradient pairings to capture the full coupling pattern. This is an innovative contribution, as structural-functional coupling has rarely been investigated across multiple dimensions. Below, I outline several concerns that should be addressed.

1. The manuscript claims that "structural gradients primarily reflected the topology of long-range white matter pathways." However, this claim is not directly supported by any figure. The authors should clarify how this interpretation is derived from their results or cite relevant references to support the statement. It would also be helpful to provide validation using existing structural (non-functional) atlases (such as the Desikan-Killiany atlas, which is already used in this paper) by comparing FCG1 and SCG1 scores across anatomical parcellations. For instance, Figure 2b shows that SCG1 exhibits much greater variability across subjects within each functional network compared to FCG1, suggesting that FCG1 is more consistent across individuals within these networks. If SCG1 scores show lower variance within anatomical parcellations, this would lend support to the interpretation that structural gradients reflect stable topological features of brain anatomy.

2. In Figure 2, the authors describe the adult structural gradients as exhibiting greater bilateral symmetry and spatial precision compared to the child group, which is supported by the visualizations. However, the boxplots for adult SCG1 and SCG2 in Figure 2b suggest greater inter-subject variability in gradient scores compared to those in the child group (Figure 2h). This apparent dissociation between spatial refinement and increased variability is intriguing from a developmental

perspective. I recommend that the authors include a brief discussion of this point, as it would strengthen the interpretation of age-related changes in structural gradient organization. Additionally, please provide details about the boxplots. For example, do the boxes represent subjects, and what do the centerline and box boundaries indicate?

3. In several instances, the manuscript blends discussions with descriptions of results. Here are two examples: "This pattern was consistent with prior findings of reduced structure-function correspondence in transmodal regions [21]" and "This aligned with previous findings of sex-related differences in SF coupling patterns [26]." For clarity, I recommend separating the presentation of results from their discussion.

4. In the behavioral prediction model, it remains unclear what the input features were, specifically, whether the subnetwork-level coupling features were used jointly in a multivariate prediction model or entered independently for each subnetwork. The kernel ridge regression (KRR) method is described using a single feature vector x_i , but the manuscript does not clarify whether this x_i represents a concatenated 400×7 feature vector (aggregating coupling features across all seven subnetworks) or whether each subnetwork was modeled separately. While Figures 3d–g display network-specific feature importance bars, suggesting that predictions may have been conducted separately for each subnetwork, Figure 3a includes only one orange box to represent prediction performance for seven networks, which is confusing. I recommend that the authors clarify how the input features were constructed and how the prediction model was implemented for both macro- and subnetwork-level coupling.

5. The sentence in the results stating that "heritability estimates decreased monotonically with increasing gradient coupling" appears inconsistent with the directionality shown in Figures 4a and 4b (and related panels). Specifically, the "average absolute values" in both 4a and 4b exhibit a decreasing trend as coupling weakens, suggesting that stronger coupling is associated with higher heritability estimates, not lower. I'm not sure if I may have missed something, so I kindly ask the authors to clarify this point.

6. The manuscript states that "individual gradients were aligned by constructing a cohort-level template and applying Procrustes rotation." I assume the correction has been done within each cohort. However, it is unclear whether gradients were aligned across the child and adult groups prior to comparison. Since gradient directions can arbitrarily flip across groups in diffusion embedding, direct group comparisons may be confounded without proper alignment.

7. While the study compares findings across the ABCD and HCP datasets, which differ in both age range and data source, the scanning protocols and preprocessing pipelines were not consistent between the two. Although the ABCD data underwent harmonization, there is no indication that the data were harmonized across cohorts. If not, this should be acknowledged and discussed as a potential limitation. Additionally, the preprocessing procedures (particularly for the ABCD dataset) are not described in sufficient detail. I recommend that the authors provide greater methodological transparency regarding the imaging protocols and preprocessing steps.

8. In the main text, the authors refer to "psychiatric symptoms" when describing associations with subnetwork-level coupling, whereas Figure 3a labels this category as "Mental Health." For consistency and clarity, I recommend using the same terminology throughout the manuscript.

9. Throughout the manuscript, the authors primarily focus on the coupling between the first gradients (FCG1 vs. SCG1). However, no explicit rationale is provided for prioritizing these gradients over higher-order ones. The authors do mention (p.7): "To illustrate this coupling heterogeneity, we focused on two representative gradient pairs, FCG1:SCG1 and FCG1:SCG2, each showing strong alignment in one age group but weak alignment in the other." It would be informative to also report heritability and transcriptomics analyses based on FCG1:SCG2 coupling.

10. When evaluating "gradient coupling strength across different developmental stages," the authors compare children (ABCD) and adults (HCP) as two broad groups. It may be beneficial to further divide the child cohort into narrower age bands to better capture the developmental evolution of SF gradient coupling across childhood and adolescence. Although this may be constrained by the limited age range of the current ABCD dataset (a point that could be acknowledged as a limitation), future work could address this using the HCP Development dataset. Such an approach would offer a more fine-grained view of developmental trajectories and could also help mitigate cross-cohort harmonization concerns raised earlier.

Reviewer #4

(Remarks to the Author)

Version 1:

Reviewer comments:

Reviewer #1

(Remarks to the Author)

The authors are to be commended for their thorough revision of the manuscript, and I appreciate the inclusion of the statistical details underlying the analyses. However, I believe the text would still benefit from some additional rephrasing to improve clarity.

1. lines 107-109: "The averaged SC and FC matrices exhibited pronounced network-level modularity, providing the organizational scaffold that underlies subsequent gradient and coupling analyses (Supplementary Fig. 1a)."

It is not clear to me how Supp. Fig 1a is a good representation of "pronounced network-level modularity".

2. Lines 109-110. "Consistent with this organization, the gradients captured low-dimensional representations of these connectivity patterns, with SCGs and FCGs emphasizing distinct spatial transitions across cortical systems"

This is very unclear to me. What defines "low-dimensional representations of these connectivity patterns", and how do SCG and FCG emphasize "distinct spatial transitions across cortical systems" ?

3. Lines 100-101. " leveraging multimodal raw neuroimaging data spanning childhood (n = 7,938) to adulthood (n= 944), we constructed SC and FC for each subject from T1-weighted, diffusion, and functional magnetic resonance imaging (MRI) data"

Please rephrase to indicate that these are two different cohorts.

4. line 124. "while the second SCG exhibited greater bilateral symmetry ($p= 0.52$) and spatial precision, indicating refined network segregation."

the word "greater" suggests that there is a difference, which is not supported by the p-value. Is this a typo?

5. Figure 2. I suggest reordering the figure, so it aligns with the developmental stages. i.e., children on the left side and adults on the right side.

lines 237-251. I appreciate the information about the feature importance, but the wording appears to overstate the findings. These are numerical differences in feature importance, and I do not think phrasing it as indication of "higher involvement"/"decreased involvement" is suitable without any statistical tests that examines differences between the groups.

Minor issues-

fix typo line 168.

I don't think it necessary to state "FDR-corrected paired t-test" in the main text, standard reporting of statistics (e.g., $t = x$, p -adjusted = x) would be sufficient and would improve readability

Reviewer #2

(Remarks to the Author)

I thank the authors for their thorough response to my comments.

I have two additional minor points for clarification:

- The 2nd FC gradient produced from the HCP young adult cohort (Figure 2A) is quite different from previously published corresponding gradient in previous work using the same dataset (e.g., Margulies et al., 2016). Could the authors comment why this may be? In adults, we usually expect clearer differentiation of visual-somatomotor networks as the 2nd gradient, while this rather looks like a task-positive/task negative pattern.

- Pages 3-4: two different sample sizes are reported for the pediatric cohort (5,343 vs 7,938) – please clarify

Reviewer #3

(Remarks to the Author)

The authors have thoroughly addressed all my concerns.

I specifically commend the inclusion of the HCP-Development dataset to perform harmonization. This resolves my concern regarding the comparability of data across cohorts and significantly strengthens the validity of the results.

I have no further comments.

Reviewer #4

(Remarks to the Author)

Authors' Response to Reviewers' Comments: “Brain Functional-Structural Gradient Coupling Reflects Development, Behavior and Genetic Influences”

We sincerely thank all the reviewers for their comments and suggestions, which helped substantially in improving the manuscript. We have thoroughly addressed all the comments and provided detailed point-by-point responses in this letter. To facilitate the reading of the revised manuscript, we have also highlighted major changes to the original manuscript in **blue** color whenever applicable.

Reviewer 1

The study by Gao et al., examines how the brain structural-functional (S-F) gradient differs between two age-stratified cohorts (i.e., ABCD and HCP-YA cohorts). The results showed cohort and sex-specific differences in SF coupling, associations between SF gradient coupling and cognitive and mental health metrics, spatial overlap between SF gradient coupling and gene expression profiles from AHBA indicating enrichment for genes expressed in deep-layer excitatory neurons, and provide regional heritability estimates for the gradient coupling metrics. The question of how structural functional gradient coupling evolves during development and relates to behavioral, genetic and molecular features is important and somewhat understudied, to my knowledge. As such, strengths of the study are its comprehensiveness and utilization of multimodal MRI across two well-characterized, large datasets of different ages. The paper is, in general, clearly written. However, there are some major substantive concerns, as well as several minor concerns, regarding the literature review, claims and interpretation of the findings and some of the analyses, which diminish enthusiasm for the manuscript in its current form.

Response: We sincerely appreciate Reviewer 1 for the thoughtful review of our work and for offering encouraging and constructive feedback. In this revision, we have thoroughly considered all comments and provided a detailed, point-by-point response below.

1. *A more comprehensive description of the previous literature should be included in the introduction and discussion. Despite methodological differences, the S-F coupling gradient has been characterized and examined across several papers previously, which also include replication datasets. E.g., Development of S-F coupling gradient using cross-sectional and longitudinal data, as well as examining associations to cognitive performance (Baum et al., 2019, PNAS, doi: 10.1073/pnas.1912034117); S-F coupling gradient in young adults, with HCP as a replication dataset (Vazquez-Rodriguez et al., 2019, PNAS, doi: 10.1073/pnas.1903403116); age-related differences S-F coupling assessed across the lifespan (Esfahlani et al., 2022, Nat Com, doi: 10.1038/s41467-022-29770-y); characterization of the S-F coupling gradient with replication dataset (Popp et al., 2024, NeuroImage, <https://doi.org/10.1016/j.neuroimage.2024.120563>); S-F coupling, age-related differences and spatial overlap with gene*

expression (Feng et al., 2024, eLife, <https://doi.org/10.7554/eLife.93325.2>); discussed in a review article (Suarez et al., 2020, Trends in Cognitive Sciences, doi: 10.1016/j.tics.2020.01.008). While these articles are cited, the characterization in the introduction of the novelty of this study should be revised. Furthermore, the SNP based heritability of the S-F coupling also has been examined previously by some of the same authors using the same datasets as the current paper (Dai et al., 2024, Imaging Neuroscience, https://doi.org/10.1162/imag_a_00346). Some of these papers also examined S-F coupling changes in sensory and associations areas across childhood and adolescence (i.e., Feng) and across the lifespan (i.e., Esfahlani), which do not fit with the authors claims in the introduction (i.e., “no study has yet tested how SF gradient coupling evolves from late childhood through early adulthood”). Thus, it would be of importance to properly put these findings in the context of previous studies.

Response: We thank the reviewer for pointing out these important prior studies. We have now expanded both the **Introduction** (lines 30–35, 55–60, 63–65) and **Discussion** (lines 355–358) sections to provide a more comprehensive overview of the literature on structure-function (SF) coupling. Specifically, we now summarize key findings from work characterizing SF coupling and its developmental or lifespan variation (Baum et al., 2020; Feng et al., 2024; Popp et al., 2024; Vázquez-Rodríguez et al., 2019; Zamani Esfahlani et al., 2022), as well as the broader conceptual review by Suárez et al., 2020 and the SNP-based heritability analysis of SF coupling by Dai et al., 2024. We have also revised our description of the novelty of the present study to explicitly acknowledge these prior contributions.

While these studies have significantly advanced our understanding of how structural and functional connectivity are aligned across the cortex and across development, they have primarily focused on node-wise coupling or global coupling indices. In contrast, our study investigates the gradient-level coupling between structural and functional topographies, thereby characterizing SF relationships within a continuous, hierarchical gradient space and across multiple spatial scales in two developmental cohorts. We now clarify in the **Introduction** section (lines 37–52, 84–95) that our study extends this literature by (i) formally defining and quantifying structural-functional gradient coupling (SFGC) as a gradient-level endophenotype across macroscale and subnetwork resolutions, (ii) demonstrating that SFGC shows developmental differences and predicts individual variation in cognitive performance and mental health-related traits, and (iii) linking SFGC to its genetic and transcriptomic underpinnings through family-based heritability and spatially constrained imaging-transcriptomic analyses. Together, these revisions more accurately situate our work within the SF coupling literature while clearly delineating its added, gradient-level contributions.

2. *Although I find most of the analyses to be appropriate, some of the claims seem broader than what the results justify. As acknowledged by the authors, any differences between the two cohorts can also be attributed to other characteristics than age, such as sample characteristics (e.g., socioeconomic status, ethnicity, scanner effects, image quality). Thus, the paper would benefit from rephrasing words like “trajectories” and “evolved”, which gives the impression of a longitudinal analysis and causality.*

Response: We sincerely thank the Reviewer for the valuable suggestion. In the revised manuscript, we have removed or rephrased any wording that could be interpreted as implying longitudinal analysis or causal relationships, and now present these results strictly as descriptive and associational findings.

3. *As a related point, the results are often presented in a ‘big picture’ summary style (e.g., p5, ‘Their second functional gradient began to resemble the adult principal axis, indicating...’) This statement is easy to understand, but what particular analysis supported this conclusion? Likewise, on p 6 there is a statistic*

for SF gradient coupling in children, and a lower C-value for adults, but from what I can tell, there was no direct statistical comparison conducted. There is more quantitative information presented in the figures and the supplements, but even there there is a fair amount of un-evenness in the level of detail provided about what is being depicted (e.g, what conclusions should be drawn from Supp Figures 1 and 3?). More detailed, quantitative information for each of the results and figures would be helpful in evaluating the strength of the claims made.

Response:

We thank the Reviewer for highlighting the need for additional quantitative support beyond qualitative descriptions. In the revised manuscript, we have conducted additional statistical tests and hypothesis tests throughout our manuscript. All findings and discussions regarding these tests are revised and updated in the **Results** and **Discussion** sections. These tests include:

- (a) Pairwise spin-rotation permutation tests to quantify cross-cohort correspondence for all FC and SC gradients (FCG1~FCG20 and SCG1~FCG20; **Supplementary Fig. 1b, c**);
- (b) Welch's t-tests with unequal variance on the macroscale-level coupling values to compare structural-functional gradient coupling between cohort (**Supplementary Fig. 1d**);
- (c) Paired t-tests to assess the asymmetry of group-level gradients across hemispheres within each cohort;
- (d) Paired t-tests to compare SFGC-outcome association difference between macroscale SFGC and sub-network SFGC (**Fig. 2a**);
- (e) Welch's t-tests with unequal variance to compare SFGC-behavior association difference between cohorts and between sex (**Fig. 2b, c**);
- (f) Paired t-tests to compare the heritability metrics between left and right hemispheres (**Fig. 4 g, p**);
- (g) Paired t-tests to compare the heritability metrics between SFGC and individual gradients (**Supplementary Table 1**);
- (h) Power analyses to assess the detectability of heritability estimates given each cohort's family structure (**Supplementary Fig. 6**);
- (i) Spin-rotation permutation tests to adjust for the spatial autocorrelation in imaging-transcriptomic analyses (**Supplementary Table 4**).

These statistical tests provide statistical evidence for the statements such as “Their second FCG showed strong similarity to the adult primary axis...” (lines 117-119). In addition, we have provided clearer quantitative interpretation to the results and figures. Specifically, for **Supplementary Fig. 1a**, we now explicitly describe that the structural and functional connectivity matrices, averaged across subjects, exhibit pronounced network-level modularity (lines 107-109). The revised manuscript text now reads: “The averaged SC and FC matrices exhibited pronounced network-level modularity, providing the organizational scaffold that underlies subsequent gradient and coupling analyses (**Supplementary Fig. 1a**).” For **Supplementary Fig. 3**, we now clarify that these figures illustrate the consistency of the feature importance maps obtained using two distinct prediction models (kernel ridge regression and multilayer perceptrons; lines 256-258). As stated in the **Analysis robustness and stability** subsection of the **Methods** section, these figures demonstrate that the spatial patterns of feature importance are highly similar across methods, with an overall Pearson correlation of 0.75 ~ 0.80. This supports the robustness of the behavioral association results and the reliability of the proposed feature importance metrics (lines 817-822). We also provide the subnetwork SFGC importance brain maps in **Supplementary Fig. 3** to improve readability.

4. In relation to cohort differences, it would be helpful if the authors addressed the within sample correlations in more depth, as within-cohort variability may influence the brain-behavior relationships within each cohort. For instance, is there more variance in the cognitive and mental health metrics in the ABCD vs HCP-YA? If so, could the authors elaborate on the potential impact of range restrictions on the brain-behavior associations?

Response: We appreciate the Reviewer’s suggestion to examine within-cohort variability. We therefore have computed the sample standard deviation (SD) of each cognitive and mental health measure in the children (ABCD) and adult (HCP-YA) cohorts as shown in Figure 1 below. Overall, the sample SDs under each metric were of similar magnitude across cohorts. For the cognitive measures, children showed modestly higher SDs for most metrics (children/adults SD ratios $\approx 1.0-1.2$), although some composites (e.g., CogFluid, CogTotal) showed very similar or slightly higher SD in adults. For the mental health measures, the pattern was mixed: adults exhibited slightly higher SDs for Anx/Dep, With/Dep, RuleBreak, and Internal, whereas children showed somewhat higher SDs for Aggressive and External, with all differences modest.

Thus, we do not observe strong or systematic range restriction in one cohort relative to the other. In addition, our primary predictive metric is the cross-validated Pearson correlation between observed and predicted scores, which is invariant to linear rescaling of the outcome and therefore not trivially driven by differences in raw variance. Notably, predictive performance is higher in children than adults even for mental health measures where adults have equal or greater variance, indicating that the stronger prediction in the child cohort cannot be explained by simple differences in outcome variability. Instead, we interpret these findings as suggesting that SFGC features account for a larger proportion of observable individual differences in cognition and mental health during childhood than in young adulthood.

Figure 1: Line chart for the sample standard deviations of cognitive and mental health metrics across cohorts.

5. The authors state that ‘the potential of SF gradient coupling as a predictive feature for behavioral traits has not been investigated’; to that end they employ two predictive modeling approaches to examine relation-

ships to mental health symptoms and cognition, but from what I can tell they only utilized cross-sectional ABCD data. Why not utilize the longitudinal data to conduct an actual predictive analysis?

Response: We appreciate the Reviewer’s thoughtful comment. Our primary aim in the present study was to characterize how SFGC relates to cognition and mental health across developmental stages, rather than to model within-individual change over time. To address this question in a principled and comparable way, we focused on the ABCD baseline wave, which closely matches the cross-sectional HCP-YA cohort in data structure. This allowed us to implement parallel cross-sectional predictive pipelines in children and adults and to directly compare the strength of brain-to-behavior associations between developmental periods. Although ABCD has a longitudinal design, leveraging follow-up waves to perform prospective prediction would require a different analytic framework and a distinct set of modeling choices. In addition, the number of participants with high-quality imaging and behavioral data at later ABCD waves is currently substantially smaller and less balanced than at baseline, which would reduce power and further complicate direct comparison with the adult HCP-YA cohort.

We fully agree that testing whether baseline SFGC prospectively predicts subsequent cognitive and mental health trajectories represents an important and complementary next step. We view the current work as a necessary first step in establishing cross-sectional predictive validity and developmental differences in SFGC-to-behavior associations, and we plan to extend this framework to longitudinal prediction in future work once sufficiently large and well-characterized multi-wave samples are available under a common processing and QC scheme. We have recognized this as a future extension in the **Discussion** section (lines 441-446).

- 6. The ABCD dataset includes MRI data across >20 sites, and it is unclear to me how the authors addressed potential differences across scanner sites. Examination of how SF coupling may differ across the scanner sites can provide important insight into the potential confounding variables within the ABCD cohort. In addition, younger children also tend to move more in the scanner, which can also influence data quality. Thus, it would be helpful if the authors compared the image quality between the two cohorts (e.g., through Euler number/SNR/tSNR/framewise displacement) to rule out the potential impact of image quality on SF coupling.*

Response: We appreciate the Reviewer for raising these important concerns regarding scanner/site effects and potential image-quality confounds. In the revised manuscript, we have implemented a comprehensive quality-control and harmonization pipeline for both gradients and SFGC before performing any analysis.

Specifically, because direct comparisons between ABCD and HCP-YA can be confounded by differences in acquisition protocols, preprocessing pipelines, and multi-site versus single-site sampling, we first introduced the Human Connectome Project-Development (HCP-D) dataset (Somerville et al., 2018) as a bridging cohort. HCP-D is a companion study to HCP-YA that extends the HCP framework to participants aged 5-21 years, uses imaging protocols and processing pipelines closely aligned with those of HCP-YA, and samples an age range that overlaps with ABCD. For these reasons, HCP-D provides a principled bridge that allows us to separate study- and pipeline-related effects from genuine developmental differences when comparing ABCD and HCP-YA.

On this basis, our harmonization procedure proceeds in two stages. First, we applied strict motion quality control by excluding participants with large mean framewise displacement ($\text{mean_FD} > 0.3\text{mm}$), yielding final samples of 5343 ABCD, 93 HCP-D, and 875 HCP-YA participants. Second, we performed ComBat-based harmonization (Johnson, Li, and Rabinovic, 2007) using the `sva` R package (Leek et al., 2012).

Within cohorts, ABCD data were harmonized for scanner and site effects, and HCP-D data were harmonized for site effects only; no scanner or site harmonization was required for the single-site HCP-YA cohort. In all cases, scanner/site were modeled as batch variables and age and sex were included as covariates, thereby preserving age- and sex-related variation while removing scanner and site differences.

To address remaining study- and pipeline-related differences between ABCD and HCP-YA, we then aligned ABCD to HCP-YA through HCP-D. Specifically, we performed ComBat harmonization with study indicator (ABCD vs. HCP-D) as the batch effect and age and sex as covariates, which allowed us to estimate the mean study-effect difference between ABCD and HCP-D, denoted as $\Delta_{\text{ABCD-HCP}}$. We then applied this mean study effect $\Delta_{\text{ABCD-HCP}}$ to each individual subject from HCP-YA, thereby placing ABCD and HCP-YA in a comparable SFGC space while preserving inherent developmental differences rather than pipeline artifacts. As shown in **Supplementary Fig. 5**, this strategy substantially improves harmonization quality, particularly by mitigating pronounced motion- and scanner-related effects in the ABCD cohort.

For consistency, we applied this harmonization procedure before using gradients, macroscale SFGC, or subnetwork SFGC in any downstream analyses, and we reran all analyses with these harmonized measures, with updated results reported in the revised manuscript. The main findings remain robust: SFGC continues to emerge as a predictive and heritable endophenotype that reflects developmental, behavioral, and genetic influences, and the key cross-cohort differences are not explained by scanner, site, or motion effects. All of these quality-control and harmonization steps, together with more explicit descriptions of preprocessing and motion exclusion criteria, are now included in the **Imaging processing and connectivity construction** and **Data harmonization** subsections of the **Methods** section.

7. *How did the authors address potential type I errors due to spatial autocorrelation between the brain maps and transcriptomic maps? The wording of the transcriptomics analyses reflects a reverse inference logic, i.e. “Transcriptomic analyses further demonstrated that highly heritable coupling patterns are enriched for genes expressed in deep-layer excitatory neurons” “Transcriptomic signatures highlight the cellular basis of SF gradient coupling heritability”, “To identify specific molecular contributors”, “To explore the cellular basis of heritability variation in SF gradient coupling”, “transcriptomic analyses revealed that the heritability of gradient coupling was enriched for excitatory neuronal signatures” this type of phrasing could be perceived as too strong given the data; a more neutral tone might be more appropriate (e.g., emphasizing the spatial overlap with gene expression maps that are enriched for genes that are highly expressed in deep layer neurons). In addition, since the gene expression values derived from AHBA are based on post-mortem data from adults, it is unclear to me how this would be an appropriate analysis to perform on brain maps derived from children. In addition, this section suffers from the same issues noted above regarding broad statements and lack of precision, e.g. “ Particularly strong signals were observed for excitatory neurons 3 and 4..” What is considered the threshold for ‘particularly strong’? was that determined a priori ?*

Response: We thank the Reviewer for these thoughtful comments regarding our imaging transcriptomic analyses. In the revised manuscript, we have made the following targeted changes to address all these concerns thoroughly.

First, to control type I error due to spatial autocorrelation, we now explicitly evaluate all imaging transcriptomic associations against spatially constrained null models. Specifically, for each regional SFGC heritability map we generate 10,000 spin-rotation permutations (Váša et al., 2018) and recompute (i) gene-heritability correlations and (ii) cell-type enrichment scores under these spatially matched nulls. From these null distributions, we derive spatially adjusted p -values for both individual genes and cell-type gene

sets. All significance claims in the transcriptomic section now refer to these spatially adjusted p -values (reported in **Supplementary Table 4**), and we define enrichment formally as spatially adjusted $p < 0.05$ (lines 794-799).

Second, we have revised the imaging-transcriptomic analyses related to the ABCD cohort to use developmentally appropriate gene-expression resources. We agree that applying adult AHBA data to children is suboptimal. In the revised analyses, AHBA is now used only for the adult (HCP-YA) cohort. For the child cohort, we instead use developmentally matched adolescent gene-expression data from the BrainSpan atlas together with age-appropriate adolescent single-cell references. This change directly addresses the concern about the suitability of transcriptomic data for pediatric imaging maps.

Finally, we have systematically softened and clarified the wording of the transcriptomic results to avoid reverse inference or mechanistic interpretations. We now describe our findings in terms of spatial correspondence between regional SFGC heritability maps and gene-expression patterns that are enriched for specific cell-type-specific gene sets, and we support these statements with explicit spatially adjusted p -values rather than purely qualitative descriptions.

8. *The study lacks information about the power analyses for the heritability estimates, which will be important given the sample size differences in the two cohorts.*

Response: We thank the Reviewer for this important suggestion. We have now performed a power analysis to evaluate the detectability of heritability under varying true heritability levels and proportions of shared environmental effects, given the relatedness structure of our cohort. Specifically, we simulated phenotypes with a grid of assumed true heritability values $h^2 = 0.01 - 0.80$ and varying proportions of shared environmental variance (0, 0.3, 0.5, 0.8), using the same relatedness matrix and family structure as in the empirical data. For each setting, we estimated heritability using the same ACE model and assessed power as the proportion of 5,000 simulations where heritability was detected at $p < 0.05$. The results (**Supplementary Fig. 6**) show that, given our current sample size and family structure, the power to detect heritability exceeds 0.8 when the true $h^2 \geq 0.2$, particularly when the proportion of shared environment is moderate to high. We focused on varying h^2 while holding the sample size fixed, as subsampling would reduce the number of informative twin pairs and thus distort the family structure, leading to an unrealistic estimate of power. These results provide a realistic characterization of our detectable effect sizes under the available cohort structure, and we have now included this power analysis in the **Heritability estimation using the ACE model** subsection of the **Methods** section.

9. *The authors state that they performed two-sample t-tests, but it is unclear if they are assuming equal or unequal variance between the groups.*

Response: We appreciate the Reviewer's request for clarification. For all comparisons between independent groups, we used Welch's two-sample t-test, which assumes unequal variances and does not impose homogeneity of variance across groups. This is the test we applied when comparing macroscale SFGC between cohorts and when assessing differences in SFGC-behavior associations across cohorts and between sexes. In contrast, for within-subject or within-region comparisons, including comparing SFGC-outcome associations between macroscale and subnetwork SFGC, comparing heritability estimates between left and right hemispheres and comparing heritability estimates between SFGC and individual gradients (SGC or FGC), we used paired t-tests, which doesn't assume equal variances. In the revised manuscript, we have now explicitly specified the exact type of t-test used in each comparison and its variance assumptions.

10. *It would be helpful if the authors provide the age-range of the final analytic samples and a demographic table stratified by sex.*

Response: We thank the reviewer for this helpful suggestion. We have now reported the age range of the final analytic samples in the **Methods** section (lines 467-468, 484) and added **Supplementary Table 3**, which summarizes demographic variables as well as cognitive and mental health measures stratified by sex and cohort.

11. *“Our approaches and results revealed that SF gradient coupling undergoes systematic neurobiological refinement from late childhood to early adulthood with distinct sex-specific trajectories.” This statement could benefit from more careful wording as it gives the impression of causality and longitudinal analyses.*

Response: Thank you for this comment. To avoid potential confusion, we have revised the statement to: “Our findings suggest age-associated changes in SFGC, with observable differences between male and female groups.” (lines 84-85).

12. *‘Finally, we integrated transcriptomic data from the AHBA to reveal that heritable gradient coupling patterns are enriched for genes expressed in deep-layer excitatory neurons’ The authors should clarify that these analyses examine the spatial overlap between brain and transcriptomic maps.*

Response: Thank you for this insightful comment. We have revised the statement and updated the relevant sections of the **Introduction**, **Results** and **Methods** sections to clarify that our imaging transcriptomic analyses assess the spatial overlap between regional heritability patterns of SFGC and age-appropriate transcriptomic maps.

13. *‘Rather than characterizing the brain as a set of discrete networks, gradient-based approaches project connectivity patterns into low-dimensional manifolds that reveal continuous axes of organization.’ It is not clear to me what the authors mean by ‘low-dimensional manifolds’, I suggest rephrasing for clarity.*

Response: We thank the Reviewer for this helpful suggestion. To improve clarity, we have revised the sentence to avoid technical terminology. The updated text now reads: “Rather than characterizing the brain as a set of discrete networks, gradient-based approaches reduce high-dimensional connectivity data into a few continuous axes that capture gradual transitions in the brain organization (Tian et al., 2020).” (lines 37-39).

14. *‘Even fewer studies have examined how gradient coupling develops over time or varies across individuals.’ It would be helpful if the authors provided references to these studies. See major comment.*

Response: Thank you for this suggestion. As responded in Question 1, we have now expanded both the **Introduction** and **Discussion** sections to provide a more comprehensive overview of the literature on structure-function (SF) coupling with proper citations. We have also revised our description of the novelty of the present study to explicitly acknowledge these prior contributions.

15. *‘Although many aspects of brain development unfold gradually across childhood and adolescence, most studies of SF coupling have focused on either mature adult brains or aggregated developmental trajectories.’ Could the authors clarify on what they mean by “aggregated developmental trajectories”? It is also hard to evaluate this claim without references.*

Response: We thank the Reviewer for this helpful comment. In the revised manuscript, we have removed the ambiguous phrase “aggregated developmental trajectories” and replaced it with a clearer and more accurate summary of prior work on SF coupling across different developmental cohorts. Specifically, we now cite and summarize studies spanning adult cohorts, infant cohorts, youth cohorts, and broader developmental samples, including those that have explicitly examined age-related differences in SF coupling (lines 30–35, 55–60). These revisions provide precise references and eliminate terminology that could be interpreted as implying longitudinal change.

16. *‘Specifically, we first thresholded the SC and FC matrices to retain the top 10% of connections, then transformed them into cosine similarity affinity matrices.’ Could the authors elaborate on how they define the top 10% connections?*

Response: We thank the Reviewer for this question. Following previous gradient-based imaging studies (Dong, Margulies, et al., 2021; Dong, X.-H. Zhang, et al., 2024; Katsumi et al., 2023; Margulies et al., 2016; Vos de Wael et al., 2020; Yeo et al., 2011), we applied proportional thresholding at the subject level: for each individual SC and FC matrix, we ranked all non-diagonal connection weights and retained the strongest 10% of edges, setting the remaining 90% to zero. This procedure controls network sparsity and reduces the influence of weak or noisy edges while maintaining comparability across participants and modalities. We then transformed these sparsified matrices into cosine similarity affinity matrices for gradient estimation. The resulting principal gradients in both the ABCD and HCP-YA cohorts closely resemble previously reported macroscale gradient topographies (Dong, X.-H. Zhang, et al., 2024; Y. Hong et al., 2025; Paquola et al., 2019; Vos de Wael et al., 2020), supporting that our findings reflect robust large-scale organizational patterns. We have clarified this thresholding procedure in the revised **Functional and structural gradient construction** subsection (lines 615-621) of the **Methods** section.

17. *‘We further observed that the SF gradient coupling strength between the principal functional gradient and the second structural gradient (FCG1:SCG2) increased from near-zero levels in children to a robust level in adults’ It would be helpful if the authors provide references to the figure or table that support this claim.*

Response: We thank the Reviewer for this helpful comment. After performing additional data harmonization during the revision, the previously observed increase in SFGC strength between the first functional gradient and the second structural gradient (FCG1:SCG2) was no longer evident. We have therefore removed this statement from the revised manuscript to ensure that the description accurately reflects the updated results.

18. *‘Furthermore, the association between SF gradient coupling and outcomes varied by scale and age cohort: in adults, subnetwork-level SF gradient coupling showed stronger association with psychiatric symptoms than macroscale gradient coupling, whereas in children, macroscale SF gradient coupling yielded exhibited associations.’ It would be helpful if the authors provided a reference to a figure or numbers that*

supports this claim. It is also unclear to me which statistical test and results that was used to support the claim of a ‘stronger association’. Also, it is unclear to me what is meant by “yielded exhibited associations” ?

Response: We thank the reviewer for pointing out these issues. We now report the averaged out-of-sample association between each observed behavioral outcome and the SFGC-predicted behavioral outcome across 100 data-splitting replicates, as summarized in **Figure 3a**, to quantify the strength of these associations. The detailed definitions of out-of-sample associations measured by the Pearson correlation coefficient (r) for continuous outcome variables and the area under the receiver operating characteristic curve (AUC) for binary outcome variables have been added to the **Model performance and feature importance** subsection of the **Method** section. In children, macroscale-/subnetwork-level SFGC shows average correlations of $r = 0.09/0.11$ with mental health outcomes, whereas in young adults the corresponding associations are slightly weaker (average $r = 0.04/0.11$). For cognitive outcomes, SFGC-cognition associations are also stronger in children on average (macroscale/subnetwork: $r = 0.26/0.30$), particularly for integrative scores such as CogTotal (macroscale/subnetwork: $r = 0.38/0.44$).

In addition, we now provide FDR-corrected paired t -tests comparing macroscale- and subnetwork-level SFGC correlations for each outcome, also shown in **Figure 3a**, to formally support the claim that subnetwork-level SFGC often shows stronger associations than macroscale SFGC.

Finally, we have removed the unclear phrase “yielded exhibited associations” and rephrased the relevant sentences to refer explicitly to these correlation estimates and statistical tests.

19. *For SF gradient coupling and cognitive performance scores, the authors state that associations get weaker in adults, but I struggle to find the statistical test was performed to support this claim other than a numerically lower correlation. In addition, the wording gives the impression that differences in correlations are due to age (e.g., “became weaker in adults”) - if there is a significant difference, a more accurate description would be “lower in the adult cohort”.*

Response: We appreciate the Reviewer raising these points. In the revised manuscript, we now explicitly state that cohort differences in SFGC-outcome associations are tested using two-sample Welch’s t -tests with FDR-corrected p -values reported in the **Results** section and **Figure 3**. We have also revised the wording to avoid implying longitudinal change; the text now states that “SFGC was in general more associated with mental health outcomes (Aggressive, External) in children (adjusted $p < 0.01$).”

20. *‘After rigorous quality control and preprocessing of the neuroimaging data, and exclusion of participants with incomplete covariate information, the final analytic sample consisted of 7,025 subjects (mean age = 9.9 years; 48.8% male).’ ‘Following quality control and data aggregation, our final analytic sample comprised 913 participants (mean age = 28.7 years; 46.55% male) from 414 families, including 111 MZ twin pairs, 59 DZ twin pairs, 511 full sibling pairs, and 95 singletons.’ It would be helpful if the authors could elaborate on the quality control procedures performed on each cohort.*

Response: We thank the Reviewer for this helpful comment. The detailed quality-control and harmonization steps for each cohort are now included in the **Imaging processing and connectivity construction** and **Data harmonization** subsections of the **Methods** section.

21. *Will the authors provide the regional values for the SF coupling gradients? This could be further used by other researchers to examine the spatial overlap with SF coupling with other cortical maps.*

Response: Thanks for this helpful suggestion. We have now provided the individual-level subnetwork SFGC for both cohorts on our GitHub repository to facilitate further spatial analyses by other researchers. Please check our full codebase (<https://github.com/Zhao-team/SF-Gradient-Coupling>).

22. *The GitHub link doesn't work. It would be helpful if the authors could open the link for review.*

Response: We thank the Reviewer for flagging this. We have corrected the link and made our full codebase publicly available on GitHub (<https://github.com/Zhao-team/SF-Gradient-Coupling>).

23. *p. 3- 'when brain organizations are more stable and specialized'- change to 'when brain organization IS more stable and specialized'*

Response: Thanks for the suggestion. The relevant paragraph has now been rewritten in the revised manuscript, and the specific wording noted no longer appears.

References

- Baum, Graham L. et al. (2020). "Development of structure–function coupling in human brain networks during youth". In: *Proceedings of the National Academy of Sciences* 117.1, pp. 771–778.
- Dai, Wei et al. (2024). "Heritability and genetic contribution analysis of structural-functional coupling in human brain". In: *Imaging Neuroscience* 2, pp. 1–19.
- Dong, Hao-Ming, Daniel S Margulies, et al. (2021). "Shifting gradients of macroscale cortical organization mark the transition from childhood to adolescence". In: *Proceedings of the National Academy of Sciences* 118.28, e2024448118.
- Dong, Hao-Ming, Xi-Han Zhang, et al. (2024). "Ventral attention network connectivity is linked to cortical maturation and cognitive ability in childhood". In: *Nature Neuroscience*, pp. 1–12.
- Feng, Guozheng et al. (2024). "Spatial and temporal pattern of structure–function coupling of human brain connectome with development". In: *eLife* 13, RP93325.
- Hong, Yoonmi et al. (2025). "Structural connectome gradients and their relationship to IQ in childhood". In: *Frontiers in Human Neuroscience* 19.
- Johnson, W Evan, Cheng Li, and Ariel Rabinovic (2007). "Adjusting batch effects in microarray expression data using empirical Bayes methods". In: *Biostatistics* 8.1, pp. 118–127.
- Katsumi, Yuta et al. (2023). "Correspondence of functional connectivity gradients across human isocortex, cerebellum, and hippocampus". In: *Communications Biology* 6.1, p. 401.
- Leek, Jeffrey T et al. (2012). "The sva package for removing batch effects and other unwanted variation in high-throughput experiments". In: *Bioinformatics* 28.6, pp. 882–883.
- Margulies, Daniel S et al. (2016). "Situating the default-mode network along a principal gradient of macroscale cortical organization". In: *Proceedings of the National Academy of Sciences* 113.44, pp. 12574–12579.
- Paquola, Casey et al. (2019). "Microstructural and functional gradients are increasingly dissociated in transmodal cortices". In: *PLoS Biology* 17.5, e3000284.

- Popp, Johanna L et al. (2024). “Structural-functional brain network coupling predicts human cognitive ability”. In: *NeuroImage* 290, p. 120563.
- Somerville, Leah H et al. (2018). “The Lifespan Human Connectome Project in Development: A large-scale study of brain connectivity development in 5–21 year olds”. In: *Neuroimage* 183, pp. 456–468.
- Suárez, Laura E et al. (2020). “Linking structure and function in macroscale brain networks”. In: *Trends in Cognitive Sciences* 24.4, pp. 302–315.
- Tian, Ye et al. (2020). “Topographic organization of the human subcortex unveiled with functional connectivity gradients”. In: *Nature Neuroscience* 23.11, pp. 1421–1432.
- Váša, František et al. (2018). “Adolescent tuning of association cortex in human structural brain networks”. In: *Cerebral Cortex* 28.1, pp. 281–294.
- Vázquez-Rodríguez, Bertha et al. (2019). “Gradients of structure–function tethering across neocortex”. In: *Proceedings of the National Academy of Sciences* 116.42, pp. 21219–21227.
- Vos de Wael, Reinder et al. (Mar. 2020). “BrainSpace: a Toolbox for the Analysis of Macroscale Gradients in Neuroimaging and Connectomics Datasets”. In: *Communications Biology* 3.1, pp. 1–10. ISSN: 2399-3642. DOI: 10.1038/s42003-020-0794-7. (Visited on 05/02/2023).
- Yeo, BT Thomas et al. (2011). “The organization of the human cerebral cortex estimated by intrinsic functional connectivity”. In: *Journal of Neurophysiology*.
- Zamani Esfahlani, Farnaz et al. (2022). “Local structure-function relationships in human brain networks across the lifespan”. In: *Nature Communications* 13.1, p. 2053.

Reviewer 2

This paper measures the similarity of functional and structural gradients (SF gradient coupling) in a cohort of children (9-10 years old, ABCD dataset) vs adults (22-35 years old, HCP dataset). The authors assess cohort-level differences in each gradient modality and their coupling across cohorts, with coupling evaluated across all regions (“macroscale”) and stratified within specific subnetworks. They show different SF coupling patterns between children and adults spanning several subnetworks. A series of within cohort analyses link SF coupling patterns to mental health and cognitive measures, identifying different magnitude and patterns of associations across cohorts and additionally uncovering sex-specific effects. More specifically, SF coupling and cognitive measure associations were generally stronger in children, while mental health relationships were stronger in adults. Heritability analyses across modality-specific and SF coupling gradients captured overall similar patterns in both groups, with slightly higher heritability estimates in children, particularly for the SF coupling assessment. Transcriptomic decoding of SF coupling heritability estimates showed associations with the expression of excitatory neurons subtypes. The paper includes an impressive breadth of analyses. However, I have several major concerns, notably regarding the motivation of the study, the way developmental comparisons were implemented, and note that several interpretations and conclusions are not supported by adequate statistical tests.

Response: We appreciate Reviewer 2’s positive feedback and insightful suggestions. We have carefully addressed all the comments in the revised manuscript. Our detailed, point-by-point responses are provided below.

1. *The introduction lacks a clear rationale for why gradients and SF gradient coupling are used and important for neurodevelopmental inquiry. Currently, the paper is framed as if the primary motivation for looking at SF gradient coupling is that it has not been done before, and therefore behavioural and genetic associations of SF coupling are unknown. What insights can SF gradient coupling bring beyond previously used measures of FC-SC coupling? What would diverging results from connectome-level measures indicate? Without this justification, it is difficult to contextualize the importance or new insights brought by the multitude of analyses performed here.*

Response: We thank the Reviewer for this insightful comment. In the revised **Introduction** section (lines 37-52), we have substantially clarified why we use gradients and structural-functional gradient coupling (SFGC), and what they add beyond more conventional SF coupling measures.

First, we now more clearly explain what gradients are and why they are relevant for development: gradient-based representations summarize whole-brain organization along a small number of continuous axes (for example, from sensory/motor regions toward association cortex). This provides a compact description of large-scale organization that is well suited for studying how brain organization changes with age.

Second, we explicitly clarify how SFGC differs from traditional SF coupling measures. Node-wise or edge-wise SF coupling focuses on how similar structural and functional connectivity are for each region or connection. In contrast, SFGC measures how structural and functional gradients align with each other across the cortex, at both macroscale and subnetwork levels. In the revised text, we state that SFGC therefore captures the correspondence between structural and functional hierarchies, rather than only local SC-FC similarity.

Third, we now briefly explain what it would mean if results based on SFGC differ from those based on node-wise or connectome-level SF coupling. If SFGC shows age or behavioral effects where node-wise SF coupling does not, this points to changes in the alignment of large-scale hierarchies rather than changes in

local coupling alone. If the opposite pattern occurs, this would suggest more local reconfiguration without major changes in the global gradient structure. We highlight that this is the main reason we complement existing SF coupling measures with a gradient-level coupling measure.

These additions make the rationale for using SFGC explicit and connect it directly to the questions about development, behavior, and genetics that we address in the remainder of the paper.

2. *The authors perform group-level comparisons of children (9-10) vs young adults (22-35) cohorts. First, as data are not harmonized across sites with methods such as Combat, it is unclear if reported effects are attributable to differences in age or general site/scanner differences. Second, and as the authors know, both white matter and functional networks change throughout the lifespan, and derivative metrics from these networks peak at different ages for different subnetworks. Childhood is a developmental window of rapid change, and even in the adult time window investigated here, FC and SC are not static (Yeatman et al., 2014, Nat Comms; Fjell et al., 2017, HBM; Betzel et al., 2014, NIMG). Most developmental studies model age effects continuously for this reason. As a result, I am not convinced that a cohort-level comparison performed on unharmonized data is appropriate.*

Response: We truly appreciate the Reviewer for these important comments. In the revised manuscript, we have performed a comprehensive data harmonization procedure to strengthen the validity of our cross-cohort comparisons and clarified our developmental framing.

First, regarding harmonization, we have implemented a ComBat-based harmonization pipeline for both gradients and SFGC and rerun all analyses using the harmonized measures. Briefly, we (i) applied strict motion quality control (excluding participants with mean framewise displacement > 0.3 mm), (ii) harmonized ABCD for scanner and site effects and HCP-Development (HCP-D) for site effects (with age and sex as covariates), and (iii) used HCP-D as a bridge to estimate and correct study/pipeline differences between ABCD and HCP-YA, thereby placing the cohorts in a comparable SFGC space while preserving age-related variation. As shown in **Supplementary Fig. 5**, this procedure substantially reduces scanner-, site-, and motion-related structure, particularly in ABCD, while retaining the cohort and age patterns central to our developmental contrasts. These procedures are now described in detail in the **Data harmonization** subsection of the **Methods** section, and all cross-cohort results in the revised manuscript are based on the harmonized data. The updated analyses continue to support our original conclusions that SFGC is a predictive and heritable endophenotype reflecting developmental, behavioral, and genetic influences.

Second, regarding the developmental framing, we fully agree that white-matter and functional networks change continuously across the lifespan and that neither childhood nor adulthood is static. Our design uses two cross-sectional cohorts (children in ABCD and young adults in HCP-YA), and we now state more clearly that our comparisons provide broad contrasts between late childhood/early adolescence and early adulthood, rather than a detailed continuous trajectory. In the revised text, we (i) adjusted the language in the **Introduction** and **Discussion** sections to avoid implying longitudinal “trajectories”, (ii) clarified that age is modeled within cohorts to account for within-group age variability, and (iii) explicitly acknowledged in the **Discussion** section that our cross-sectional, two-cohort design cannot capture nonlinear or region-specific age effects across the full lifespan, and that a reliable characterization of such effects will require densely sampled longitudinal data.

Together, these changes ensure that our cross-cohort comparisons are based on harmonized, site-corrected SFGC measures and that our developmental interpretations are framed as broad cross-sectional contrasts with explicit acknowledgment of the limits of our design.

3. *Several analytical choices are not adequately justified. For example, some analyses focus only on G1 and G2, the coupling of FCG1 and SCG1 or FCG1 and SCG2, and the heritability analyses look at 20 distinct gradients. It is hard to make sense of these analytical variations and interpret findings coherently. For example, would we expect different downstream findings from measuring SF gradient coupling using FCG1:SCG1 vs FCG1:SCG2?*

Response: We thank the Reviewer for raising this important point. In the revised manuscript, we now make it explicit that all primary analyses, including cross-cohort comparisons, behavioral associations, and heritability analyses, are based on SFGC indices derived from the full set of 20×20 functional-structural gradient combinations rather than on any single gradient pair. The place we introduce an individual pair as a demonstration is when illustrating how SFGC manifests within Yeo-7 and Desikan-Killiany subnetworks. In that context, we focus on a representative pair, FCG1:SCG1, because the principal gradients explain the largest proportion of variance and show different patterns in the canonical somatomotor-visual hierarchy across cohorts, making them the most interpretable for demonstrating heterogeneity across networks. This example is used to show that some subnetworks (e.g., sensory/motor) remain strongly coupled while others (e.g., DMN, control networks) are more decoupled, in line with prior work. It does not define or constrain the SFGC measures used in downstream analyses.

Regarding the specific question of FCG1:SCG1 vs. FCG1:SCG2, different gradient pairs can indeed show different child-to-adult contrasts at the pairwise level, because they reflect different axes of organization. However, our conclusions about behavioral associations and heritability across cohorts are drawn from SFGC measures computed from the full gradient set, not from any single pair. We have clarified this distinction in the **Methods** section and in the relevant figure and text in the revised manuscript.

4. *Many claims in the paper are not statistically supported. For example, differences between young and adult cohorts on FC and SC gradients are not quantified, only described qualitatively. As a result, the effect of modality-specific differences on SF gradient coupling differences cannot be assessed (i.e. are the effects driven by one modality over the other). On pages 6-8, the authors claim “Together, these findings established macroscale and subnetwork-level SF gradient coupling as two equally informative metrics for quantifying correspondence between brain structural and functional organization.” Yet, no quantitative comparison of the two approaches is performed - how can they be qualified as equally informative? In the heritability analyses, the authors report “These findings suggested that SF gradient coupling captured highly heritable multimodal axes of brain organization that were not fully explained by either modality alone.”. They support these claims by simply reporting heritability metrics FC, SC and SF coupling gradients, yet there are no models implemented to assess unique variance explained or direct statistical comparisons of these heritability metrics.*

Response: We appreciate the Reviewer’s comment. In the revised manuscript, we have conducted additional statistical tests and hypothesis tests throughout our manuscript. All findings and discussions regarding these tests are revised and updated in the **Results** and **Discussion** sections. These tests include:

- (a) Pairwise spin-rotation permutation tests to quantify cross-cohort correspondence for all FC and SC gradients (FCG1~FCG20 and SCG1~FCG20; **Supplementary Fig. 1b, c**);
- (b) Welch’s t-tests with unequal variance on the macroscale-level coupling values to compare structural-functional gradient coupling between cohort (**Supplementary Fig. 1d**);
- (c) Paired t-tests to assess the asymmetry of group-level gradients across hemispheres within each cohort;

- (d) Paired t-tests to compare SFGC-outcome association difference between macroscale SFGC and sub-network SFGC (**Fig. 2a**);
- (e) Welch’s t-tests with unequal variance to compare SFGC-behavior association difference between cohorts and between sex (**Fig. 2b, c**);
- (f) Paired t-tests to compare the heritability metrics between left and right hemispheres (**Fig. 4 g, p**);
- (g) Paired t-tests to compare the heritability metrics between SFGC and individual gradients (**Supplementary Table 1**);
- (h) Power analyses to assess the detectability of heritability estimates given each cohort’s family structure (**Supplementary Fig. 6**);
- (i) Spin-rotation permutation tests to adjust for the spatial autocorrelation in imaging-transcriptomic analyses (**Supplementary Table 4**).

Specifically, for differences in FC and SC gradients across cohorts, we now perform pairwise spin permutation tests (10,000 permutations) on harmonized gradients from ABCD and HCP-YA. Before comparison, gradient signs were aligned to a common reference to remove orientation ambiguity. For each ABCD gradient (FCG1~FCG20; SCG1~SCG20), we computed its spatial correlation with all 20 corresponding gradients in HCP and evaluated significance using the spin test, with FDR correction applied across the 20×20 comparisons. The resulting FDR-corrected p -value heatmaps are shown in **Supplementary Fig. 1b, c** and provide a formal basis for the qualitative descriptions in the main text.

Regarding the statement that “macroscale and subnetwork-level SFGC are two equally informative metrics,” our intention was not to claim formal equivalence but to emphasize that both scales of SFGC represent meaningful levels of organization that provide useful information about structure-function correspondence. We have revised the manuscript to clarify this wording, and the sentence reads: “Together, these findings established that both macroscale- and subnetwork-level SFGC represent meaningful levels of organization, and that examining both scales provides a more complete characterization of SF correspondence.” We now have also reported FDR-corrected paired t -tests comparing macroscale vs. subnetwork SFGC in their associations with each outcome, as shown in **Figure 3a** and described in the **Results** section.

For the heritability analyses, to statistically compare the metric under SFGC with those under unimodal gradients, we have added a dedicated analysis at matched subnetwork resolution. Because SFGC was defined at the Yeo-7 / Desikan-Killiany subnetwork level, we have further summarized vertex-wise FC and SC gradients at the same resolution, estimated heritability for each subnetwork-level phenotype using the same ACE model, and conducted two-sided paired t -tests comparing SFGC with its corresponding FC and SC gradients as presented in **Supplementary Table 1** with Bonferroni corrected p -values. This framework allows explicit statistical comparison of heritability across modalities.

5. *HCP and ABCD gradients were aligned to a “cohort-level” template. By aligning each cohort to distinct templates, does this procedure not run the risk of spuriously inflating group effects?*

Response: We appreciate the Reviewer’s thoughtful question. In our analysis, Procrustes alignment was performed only within each cohort: for ABCD and HCP-YA separately, each subject’s gradients were aligned to that cohort’s empirical mean gradient. This “template” is simply the within-cohort average of subject-level embeddings in a common anatomical space. It is not shared across cohorts and is not derived from any external reference. The Procrustes step applied an orthogonal transformation in the gradient space to resolve the arbitrary sign and orientation of the diffusion embedding. It does not change the

spatial pattern of gradient values on the cortical surface or their variance, nor does it move one cohort toward the other.

Because the ABCD and HCP-YA templates were estimated independently and alignment was restricted to within-cohort rotations, this procedure is expected to reduce within-cohort noise rather than create or inflate between-cohort differences. In fact, the risk of using a single joint template across cohorts would pose the opposite risk, by forcing the cohorts into an artificially similar embedding. Moreover, all key cross-cohort comparisons in the revised manuscript are based on harmonized gradients and SFGC measures after our ComBat-based harmonization procedure correction for scanner, site, and study/pipeline effects, as detailed in the **Data harmonization** subsection in the **Methods** section. The developmental differences we report are therefore observed after both within-cohort orientation alignment and across-cohort harmonization.

6. *The use of causally-ambiguous language in describing transcriptomic findings is unwarranted for a correlation approach such as the one applied here.*

Response: We appreciate the Reviewer for raising this comment. We have thoroughly revised the **Results** and **Methods** sections to remove causally suggestive terminology. We now describe the findings through our correlation-based analyses strictly as spatial associations or correspondence between heritability maps and gene expression patterns.

7. *The Discussion is overall very short and provides little to no contextualization of results with the literature, focusing on a summary of the findings.*

Response: Thank you for this comment. As suggested, in the revised manuscript, we have substantially expanded the **Discussion** section to better situate our findings within the existing literature and clarify their conceptual implications.

8. *The link provided to reproduce the original analyses in this paper is not functional (<https://github.com/Naomi-Ding/SF-Gradient-Coupling> - 404 not found). It is thus not possible to evaluate or review the code used to produce these findings.*

Response: We thank the Reviewer for flagging this. We have made our full codebase publicly available on GitHub (<https://github.com/Zhao-team/SF-Gradient-Coupling>) to facilitate independent verification of our analyses.

9. *As a minor point, the figures are generally quite dense, making the text and smaller figure panels difficult to parse. Also, the magnitude and topography of group differences are difficult to evaluate when adult and child effects are not presented on the same scale (e.g., Fig 1d-j; f-l, but the point holds throughout the manuscript).*

Response: Thank you for your comments. In the revised manuscript, we have revised all figures to be more visually accessible by increasing font sizes and reorganizing subfigures. When feasible, we present cohort-related effects using a common scale to facilitate evaluation of both the magnitude and spatial pattern of group differences. For selected subfigures including Fig. 2a, b, d and Fig. 2g, h, j, we retained the original scaling to allow clearer visualization of the intrinsic brain organizational hierarchies within each cohort,

which would be less discernible under a shared scale. Importantly, all across-cohort comparisons are now based on formal statistical tests rather than visual inspection alone.

Reviewer 3

This study introduces and characterizes a novel neuroimaging metric, structural-functional (SF) gradient coupling, which quantifies the alignment between structural and functional gradients both globally and within distinct brain networks. The authors examine how this coupling varies across developmental stages, differs by sex, relates to behavior and mental health, and reflects genetic and transcriptomic influences. Given that the gradients are derived separately for functional and structural modalities, and thereby their rank order does not necessarily correspond across modalities, the authors opted to explore all possible gradient pairings to capture the full coupling pattern. This is an innovative contribution, as structural-functional coupling has rarely been investigated across multiple dimensions. Below, I outline several concerns that should be addressed.

Response: We really appreciate the Referee for taking the time reviewing our paper and the positive and constructive feedback. In this revision, we have thoroughly addressed all the comments, and our point-to-point response is provided below.

1. *The manuscript claims that “structural gradients primarily reflected the topology of long-range white matter pathways.” However, this claim is not directly supported by any figure. The authors should clarify how this interpretation is derived from their results or cite relevant references to support the statement. It would also be helpful to provide validation using existing structural (non-functional) atlases (such as the Desikan-Killiany atlas, which is already used in this paper) by comparing FCG1 and SCG1 scores across anatomical parcellations. For instance, Figure 2b shows that SCG1 exhibits much greater variability across subjects within each functional network compared to FCG1, suggesting that FCG1 is more consistent across individuals within these networks. If SCG1 scores show lower variance within anatomical parcellations, this would lend support to the interpretation that structural gradients reflect stable topological features of brain anatomy.*

Response: We thank the Reviewer for pointing out this ambiguous phrasing in the original manuscript. In the revised paper, we have removed this claim and replaced it with a more neutral, empirically grounded description that reads: “*The averaged SC and FC matrices exhibited pronounced network-level modularity, providing the organizational scaffold that underlies subsequent gradient and coupling analyses (Supplementary Fig. 1a). Consistent with this organization, the gradients capture low-dimensional representations of these connectivity patterns, with SCGs and FCGs emphasizing distinct spatial transitions across cortical systems (Margulies et al., 2016; Park et al., 2021).*”.

Regarding the suggestion on comparing FCG1 and SCG1 scores under the Desikan-Killiany atlas, we first clarify that the boxplots in **Figure 2b, h** do not depict subjects’ gradient scores. Instead, they show parcel-averaged gradient values within each functional subnetwork, and therefore quantify parcelwise spatial variability rather than individual variability. Following the Reviewer’s recommendation, we examined SCG and FCG patterns with respect to Desikan-Killiany parcels. These analyses did not provide strong statistical support for the stronger claim that structural gradients “primarily” reflect long-range anatomical topology relative to functional gradients. We therefore refrain from making that claim in the revised manuscript and limit our interpretation to the observed, data-driven differences between SCGs and FCGs. The corresponding descriptions in **Supplementary Fig. 1a** and the **Results** and **Discussion** sections have been updated accordingly.

2. *In Figure 2, the authors describe the adult structural gradients as exhibiting greater bilateral symmetry and spatial precision compared to the child group, which is supported by the visualizations. However, the*

boxplots for adult SCG1 and SCG2 in Figure 2b suggest greater inter-subject variability in gradient scores compared to those in the child group (Figure 2h). This apparent dissociation between spatial refinement and increased variability is intriguing from a developmental perspective. I recommend that the authors include a brief discussion of this point, as it would strengthen the interpretation of age-related changes in structural gradient organization. Additionally, please provide details about the boxplots. For example, do the boxes represent subjects, and what do the centerline and box boundaries indicate?

Response: We thank the Reviewer for this insightful suggestion. We have now explicitly included relevant discussions on this aspect in the **Discussion** section as follows: “A related developmental feature of SCGs was the dissociation between large-scale spatial organization and local dispersion of gradient values. In adults, principal structural gradients exhibited clearer large-scale spatial organization, including more pronounced bilateral symmetry, while at the same time showing greater dispersion of gradient values within specific subnetworks. Importantly, these observations reflected distinct aspects of gradient structure: large-scale spatial organization captures the gradients layout across the cortex, whereas local dispersion reflects heterogeneity in gradient expression across vertices within subnetworks. From a developmental perspective, this apparent dissociation suggested that maturation may support more stable global organization of structural gradients while allowing increased heterogeneity in how individual vertices express these gradients.”

In addition, we have clarified the construction of the boxplots in the revised manuscript. The boxplots do not represent subject-level variability. Instead, they summarize cohort-level gradient scores across cortical vertices. These vertex-level scores were then grouped according to the Yeo-7 functional networks, and the boxplots show the distribution of vertex scores within each network. The center line corresponds to the median vertex score, the box denotes the interquartile range (IQR), and whiskers represent $1.5 \times IQR$, with points outside the whiskers plotted as outliers. This clarification has been added to the **Figure 2** caption in the revised paper.

3. In several instances, the manuscript blends discussions with descriptions of results. Here are two examples: “This pattern was consistent with prior findings of reduced structure-function correspondence in transmodal regions [21]” and “This aligned with previous findings of sex-related differences in SF coupling patterns [26].” For clarity, I recommend separating the presentation of results from their discussion.

Response: We thank the Reviewer for this helpful suggestion. In the revised manuscript, we have comprehensively reorganized the text to maintain a clearer separation between **Results** and **Discussion** sections. The **Results** section now focuses on describing empirical findings, while literature-based commentary has been moved to the **Discussion** section.

4. In the behavioral prediction model, it remains unclear what the input features were, specifically, whether the subnetwork-level coupling features were used jointly in a multivariate prediction model or entered independently for each subnetwork. The kernel ridge regression (KRR) method is described using a single feature vector x_i , but the manuscript does not clarify whether this x_i represents a concatenated 400×7 feature vector (aggregating coupling features across all seven subnetworks) or whether each subnetwork was modeled separately. While Figures 3d-g display network-specific feature importance bars, suggesting that predictions may have been conducted separately for each subnetwork, Figure 3a includes only one orange box to represent prediction performance for seven networks, which is confusing. I recommend that

the authors clarify how the input features were constructed and how the prediction model was implemented for both macro- and subnetwork-level coupling.

Response: We appreciate the Reviewer’s attention to this important technical detail. Throughout behavioral prediction analyses, each scale of SFGC features were modeled jointly in a single multivariate model. For macroscale SFGC, we fitted one prediction model per cohort, where each subject’s input feature vector consists of the full set of 400 (20×20) gradient couplings, and both KRR and MLP were trained on this joint feature space. For subnetwork-level SFGC, we concatenated the coupling features from all subnetworks into a single feature vector, which had a dimension of 5,600 (400 gradient couplings \times 14 Yeo-7 subnetworks). Similarly, each KRR and MLP model were trained on this joint 5,600-dimensional feature space. The network-specific feature-importance profiles shown in **Figure 3d-g** were derived from this single subnetwork-level model and are therefore directly comparable across subnetworks. We have clarified this modeling strategy in the **Structural-functional gradient coupling, Associations between gradient coupling and behavioral outcomes** and **Model performance and feature importance** subsections of the **Methods** section, and have modified related text for **Figure 3** in the **Results** section.

5. *The sentence in the results stating that “heritability estimates decreased monotonically with increasing gradient coupling” appears inconsistent with the directionality shown in Figures 4a and 4b (and related panels). Specifically, the “average absolute values” in both 4a and 4b exhibit a decreasing trend as coupling weakens, suggesting that stronger coupling is associated with higher heritability estimates, not lower. I’m not sure if I may have missed something, so I kindly ask the authors to clarify this point.*

Response: Thank you for catching this ambiguity and for the careful reading. The original phrasing intended to describe the ordering of coupling indices across functional and structural gradients, not the magnitude of coupling strength. Our results showed that gradient coupling pairs involving lower-order SC and FC gradients exhibit both stronger coupling and higher heritability, whereas coupling that involves higher-order gradients shows weaker coupling and lower heritability. Thus, the hierarchical decline in heritability mirrors the hierarchical decline in coupling strength across gradient orders, not the relationship between heritability and the magnitude of coupling per se.

We agree that the statement could bring confusion and have revised the text accordingly to remove this ambiguity and now explicitly state that heritability decreases systematically across couplings involving higher-order gradients, consistent with the patterns shown in **Figure 4a-d, j-m**.

6. *The manuscript states that “individual gradients were aligned by constructing a cohort-level template and applying Procrustes rotation.” I assume the correction has been done within each cohort. However, it is unclear whether gradients were aligned across the child and adult groups prior to comparison. Since gradient directions can arbitrarily flip across groups in diffusion embedding, direct group comparisons may be confounded without proper alignment.*

Response: We thank the Reviewer for this thoughtful comment. In our study, Procrustes alignment was performed separately within ABCD and HCP-YA to ensure consistent gradient orientation at the individual level within that cohort. We did not perform cross-cohort Procrustes alignment in order to avoid forcing the two cohorts into an artificially similar embedding. We agree that gradient axes from diffusion embedding can, in principle, flip across cohorts; however, this potential cross-cohort sign flip does not affect any of our downstream analyses. Specifically, (i) the cross-cohort spatial correspondence in SCGs and FCGs was

derived from the absolute value of their spatial correlation, and the cross-cohort difference in SFGC for each gradient pair was calculated based on the magnitude of the cohort-level coupling values; (ii) brain-behavior prediction was performed within each cohort, so out-of-sample correlations between predicted and observed scores are invariant to a global sign flip of SFGC; and (iii) ACE heritability analyses quantify variance components, which are also mathematically unchanged by multiplying gradient values by -1 . In addition, we have applied a comprehensive across-cohort harmonization of both gradients and SFGC to remove scanner, site, and study/pipeline effects before conducting any comparisons between ABCD and HCP-YA. The full procedure is now described in the **Data harmonization** subsection in the **Methods** section. This ensures that group differences reflect biological rather than preprocessing-related variation. Taken together, these design choices ensure that any potential arbitrary differences in gradient sign across cohorts do not confound our group comparisons or the resulting inferences.

7. *While the study compares findings across the ABCD and HCP datasets, which differ in both age range and data source, the scanning protocols and preprocessing pipelines were not consistent between the two. Although the ABCD data underwent harmonization, there is no indication that the data were harmonized across cohorts. If not, this should be acknowledged and discussed as a potential limitation. Additionally, the preprocessing procedures (particularly for the ABCD dataset) are not described in sufficient detail. I recommend that the authors provide greater methodological transparency regarding the imaging protocols and preprocessing steps.*

Response: We thank the Reviewer for these important comments on preprocessing and harmonization. In the revised manuscript, we have implemented a comprehensive quality-control and harmonization pipeline for both gradients and SFGC before performing any analyses, and we now provide more detailed descriptions of the preprocessing steps for each dataset.

Specifically, because direct comparisons between ABCD and HCP-YA can be confounded by differences in acquisition protocols, preprocessing pipelines, and multi-site versus single-site sampling, we first introduced the Human Connectome Project-Development (HCP-D) dataset (Somerville et al., 2018) as a bridging cohort. HCP-D is a companion study to HCP-YA that extends the HCP framework to participants aged 5-21 years, uses imaging protocols and processing pipelines closely aligned with those of HCP-YA, and samples an age range that overlaps with ABCD. For these reasons, HCP-D provides a principled bridge that allows us to separate study- and pipeline-related effects from genuine developmental differences when comparing ABCD and HCP-YA.

On this basis, our harmonization procedure proceeds in two stages. First, we applied strict motion quality control by excluding participants with large mean framewise displacement ($\text{mean_FD} > 0.3\text{mm}$), yielding final samples of 5343 ABCD, 93 HCP-D, and 875 HCP-YA participants. Second, we performed ComBat-based harmonization (Johnson, C. Li, and Rabinovic, 2007) using the `sva` R package (Leek et al., 2012). Within cohorts, ABCD data were harmonized for scanner and site effects, and HCP-D data were harmonized for site effects only; no scanner or site harmonization was required for the single-site HCP-YA cohort. In all cases, scanner/site were modeled as batch variables and age and sex were included as covariates, thereby preserving age- and sex-related variation while removing scanner and site differences.

To address remaining study- and pipeline-related differences between ABCD and HCP-YA, we then aligned ABCD to HCP-YA through HCP-D. Specifically, we performed ComBat harmonization with study indicator (ABCD vs. HCP-D) as the batch effect and age and sex as covariates, which allowed us to estimate the mean study-effect difference between ABCD and HCP-D, denoted as $\Delta_{\text{ABCD-HCP}}$. We then applied this mean study effect $\Delta_{\text{ABCD-HCP}}$ to each individual subject from HCP-YA, thereby placing ABCD and

HCP-YA in a comparable SFGC space while preserving inherent developmental differences rather than pipeline artifacts. As shown in **Supplementary Fig. 5**, this strategy substantially improves harmonization quality, particularly by mitigating pronounced motion- and scanner-related effects in the ABCD cohort.

For consistency, we applied this harmonization procedure before using gradients, macroscale SFGC, or subnetwork SFGC in any downstream analyses, and we reran all analyses with these harmonized measures, with updated results reported in the revised manuscript. The main findings remain robust: SFGC continues to emerge as a predictive and heritable endophenotype that reflects developmental, behavioral, and genetic influences, and the key cross-cohort differences are not explained by scanner, site, or motion effects. All of these quality-control and harmonization steps, together with more explicit descriptions of preprocessing and motion exclusion criteria, are now included in the **Imaging processing and connectivity construction** and **Data harmonization** subsections of the **Methods** section.

8. *In the main text, the authors refer to "psychiatric symptoms" when describing associations with subnetwork-level coupling, whereas Figure 3a labels this category as "Mental Health." For consistency and clarity, I recommend using the same terminology throughout the manuscript.*

Response: We thank the Reviewer for this helpful suggestion. We have revised the terminology throughout the manuscript to ensure consistency and now uniformly use the term "Mental Health" to describe this behavioral domain in both the main text and figures.

9. *Throughout the manuscript, the authors primarily focus on the coupling between the first gradients (FCG1 vs. SCG1). However, no explicit rationale is provided for prioritizing these gradients over higher-order ones. The authors do mention (p.7): "To illustrate this coupling heterogeneity, we focused on two representative gradient pairs, FCG1:SCG1 and FCG1:SCG2, each showing strong alignment in one age group but weak alignment in the other." It would be informative to also report heritability and transcriptomics analyses based on FCG1:SCG2 coupling.*

Response: We thank the Reviewer for this helpful suggestion. To clarify, all our primary analyses, including cross-cohort comparisons, behavioural associations, and heritability analyses, are based on SFGC indices derived from the full set of 20×20 functional-structural gradient combinations rather than on any single gradient pair. The place we introduce an individual pair is when illustrating how SFGC manifests within Yeo-7 and Desikan-Killiany subnetworks. In that context, we focus on a representative pair, FCG1:SCG1, because the principal gradients explain the largest proportion of variance and show different patterns in the cortical hierarchy across cohorts, making them the most interpretable for demonstrating heterogeneity across networks. This example is used to show that some subnetworks (e.g., sensory/motor) remain strongly coupled while others (e.g., DMN, control networks) are more decoupled, in line with prior work. It does not define or constrain the SFGC measures used in downstream analyses.

For the heritability analyses, as shown in **Figure 4**, we systematically quantified heritability for all 20×20 FC-SC gradient combinations, FCG i :SCG j with $i, j = 1, \dots, 20$, summarizing both the global average patterns and the spatial/subnetwork-level distribution of genetic influence across all coupling pairs. These results demonstrate a hierarchical structure in genetic contributions that decreases with higher-order gradient indices. Building on this global characterization, we now chose to only highlight the FCG1:SCG1 pair in the main text because it consistently exhibited the strongest and most spatially coherent heritable signal in both cohorts, making it the most biologically interpretable and statistically robust phenotype for downstream transcriptomic enrichment analyses.

10. When evaluating “gradient coupling strength across different developmental stages,” the authors compare children (ABCD) and adults (HCP) as two broad groups. It may be beneficial to further divide the child cohort into narrower age bands to better capture the developmental evolution of SF gradient coupling across childhood and adolescence. Although this may be constrained by the limited age range of the current ABCD dataset (a point that could be acknowledged as a limitation), future work could address this using the HCP Development dataset. Such an approach would offer a more fine-grained view of developmental trajectories and could also help mitigate cross-cohort harmonization concerns raised earlier.

Response: We thank the Reviewer for this valuable suggestion. We agree that finer age segmentation could, in principle, provide a more detailed view of developmental changes in SFGC. However, the ABCD baseline cohort used here spans a relatively narrow age range (9-10 years), so further subdividing this sample would substantially reduce power and limit the interpretability of any inferred “trajectories”. In the revised manuscript, we now explicitly acknowledge this as a limitation and clarify that our current analyses are intended to provide a broad cross-sectional contrast between late childhood and early adulthood, rather than to model continuous developmental change.

As the Reviewer notes, future work using longitudinal follow-up data and broader developmental cohorts, such as HCP-Development, will be better suited to characterizing finer-grained trajectories of SFGC across childhood and adolescence. In the revision, we actually incorporate the HCP-Development dataset into our ComBat-based harmonization procedure as a bridging cohort, which helps reduce cross-cohort confounds and better links ABCD and HCP-YA. Consistent with this, we have also revised the wording in the **Introduction** and **Discussion** sections to remove or rephrase any language that could be interpreted as implying trajectories, temporal progression, or causal relationships, and now present our cross-cohort results strictly as descriptive and associational findings.

References

- Johnson, W Evan, Cheng Li, and Ariel Rabinovic (2007). “Adjusting batch effects in microarray expression data using empirical Bayes methods”. In: *Biostatistics* 8.1, pp. 118–127.
- Leek, Jeffrey T et al. (2012). “The sva package for removing batch effects and other unwanted variation in high-throughput experiments”. In: *Bioinformatics* 28.6, pp. 882–883.
- Margulies, Daniel S et al. (2016). “Situating the default-mode network along a principal gradient of macroscale cortical organization”. In: *Proceedings of the National Academy of Sciences* 113.44, pp. 12574–12579.
- Park, Bo-yong et al. (2021). “Signal diffusion along connectome gradients and inter-hub routing differentially contribute to dynamic human brain function”. In: *Neuroimage* 224, p. 117429.
- Somerville, Leah H et al. (2018). “The Lifespan Human Connectome Project in Development: A large-scale study of brain connectivity development in 5–21 year olds”. In: *Neuroimage* 183, pp. 456–468.

Reviewer 4

Response: We sincerely thank you for your thoughtful review and valuable feedback. We have carefully addressed all the comments in the revised manuscript.

Authors' Response to Reviewers' Comments: “Brain Functional-Structural Gradient Coupling Reflects Development, Behavior and Genetic Influences”

We sincerely thank all the reviewers for their remaining comments and suggestions. We have thoroughly addressed all the them and provided detailed point-by-point responses in this letter.

Reviewer 1

The authors are to be commended for their thorough revision of the manuscript, and I appreciate the inclusion of the statistical details underlying the analyses. However, I believe the text would still benefit from some additional rephrasing to improve clarity.

Response: We sincerely appreciate Reviewer 1 for the thoughtful review of our work and for offering encouraging and constructive feedback. In this revision, we have thoroughly considered all comments and provided a detailed, point-by-point response below.

1. *lines 107-109: “The averaged SC and FC matrices exhibited pronounced network-level modularity, providing the organizational scaffold that underlies subsequent gradient and coupling analyses (Supplementary Fig. 1a).” It is not clear to me how Supp. Fig 1a is a good representation of “pronounced network-level modularity”.*

Response: We thank the reviewer for this helpful comment. We agree that Supplementary Fig. 1a provides a visual illustration of network-level structure rather than a formal quantification of modularity. To avoid overstatement and improve clarity, we have revised the sentence as follows: “The averaged SC and FC matrices displayed clear network-level structure (Supplementary Fig. 1a), forming the connectivity framework for subsequent gradient and coupling analyses.”

2. *Lines 109-110. “Consistent with this organization, the gradients captured low-dimensional representations of these connectivity patterns, with SCGs and FCGs emphasizing distinct spatial transitions across cortical systems.” This is very unclear to me. What defines “low-dimensional representations of these connectivity patterns”, and how do SCG and FCG emphasize “distinct spatial transitions across cortical systems” ?*

Response: We thank the reviewer for pointing out the lack of clarity. We agree that the original wording was too abstract. Our intention was to describe gradients as low-dimensional summaries of the structural and functional connectivity matrices obtained through diffusion embedding. To improve clarity and avoid

ambiguous phrasing, we have revised the sentence as follows: “Applying diffusion embedding to the connectivity affinity matrices yielded a small number of principal gradient axes that summarize large-scale patterns in SC and FC.”

3. *Lines 100-101. “leveraging multimodal raw neuroimaging data spanning childhood (n = 7,938) to adulthood (n= 944), we constructed SC and FC for each subject from T1-weighted, diffusion, and functional magnetic resonance imaging (MRI) data” Please rephrase to indicate that these are two different cohorts.*

Response: We thank the reviewer for this suggestion. We have revised the sentence to explicitly clarify that the childhood and adulthood samples were derived from two independent cohorts (ABCD and HCP-YA), rather than a single continuous developmental sample. Now the sentence reads: “Leveraging multimodal neuroimaging data from two independent cohorts, including children from the ABCD study and young adults from the HCP cohort, ...”.

4. *line 124. “while the second SCG exhibited greater bilateral symmetry (p= 0.52) and spatial precision, indicating refined network segregation.” the word “greater” suggests that there is a difference, which is not supported by the p-value. Is this a typo?*

Response: We thank the reviewer for this careful observation. Since hemispheric asymmetry was assessed within each cohort separately, we have revised the sentence to provide a statistically grounded description: “In adults, the first SCG similarly separated VIN and SMN, while the second SCG did not exhibit significant hemispheric asymmetry (paired t -test $p = 0.52$), displaying a bilaterally symmetric spatial pattern.”

5. *Figure 2. I suggest reordering the figure, so it aligns with the developmental stages. i.e., children on the left side and adults on the right side. lines 237-251. I appreciate the information about the feature importance, but the wording appears to overstate the findings. These are numerical differences in feature importance, and I do not think phrasing it as indication of “higher involvement”/“decreased involvement” is suitable without any statistical tests that examines differences between the groups.*

Response: We thank the reviewer for these helpful suggestions. Figure 2 has been reordered so that children are shown on the left and adults on the right, consistent with the developmental progression. In addition, we agree that the previous wording for feature importance was too strong, as these values reflect descriptive numerical differences rather than formally tested between-group effects. We have therefore revised the text to use more neutral language and avoid implying statistical significance where no direct group comparison was performed.

Minor issues:

fix typo line 168

I don't think it necessary to state “FDR-corrected paired t-test” in the main text, standard reporting of statistics (e.g., $t = x$, p -adjusted = x) would be sufficient and would improve readability

Response: We thank the reviewer for these helpful suggestions. The typo in line 168 has been corrected. Due to the various types of t tests and the existence of both FDR-correction and Bonferroni-correction in the manuscript, we find it necessary to keep the term “FDR-corrected paired t-tests” for technical clarity.

Reviewer 2

I thank the authors for their thorough response to my comments. I have two additional minor points for clarification:

Response: We appreciate Reviewer 2’s positive feedback and insightful suggestions. We have carefully addressed all the comments in the revised manuscript. Our detailed, point-by-point responses are provided below.

- *The 2nd FC gradient produced from the HCP young adult cohort (Figure 2A) is quite different from previously published corresponding gradient in previous work using the same dataset (e.g., Margulies et al., 2016). Could the authors comment why this may be? In adults, we usually expect clearer differentiation of visual-somatomotor networks as the 2nd gradient, while this rather looks like a task-positive/task negative pattern.*

Response: We thank the reviewer for this insightful observation. We agree that the second functional connectivity gradient (FCG2) derived from the HCP young adult cohort differs somewhat from the pattern reported in Margulies et al., 2016. Importantly, in our results the visual and somatomotor systems still occupy opposite ends of the gradient, consistent with the expected large-scale sensory differentiation, although additional networks appear more pronounced, giving the gradient a pattern that visually resembles a task-positive/task-negative axis.

One likely reason lies in differences in the representation of the functional connectome. While Margulies et al., 2016 derived gradients from a dense whole-brain connectome including both cortical grayordinates and subcortical voxels, our SBCI framework focuses on the cortical manifold and constructs gradients using an atlas-free cortical representation (Cole et al., 2021). Prior work has shown that gradient values can vary depending on the spatial representation of the connectome and analysis choices (Vos de Wael et al., 2020; Nenning et al., 2024).

In addition, higher-order gradients (e.g., FCG2 and FCG3) can be sensitive to small variations in connectivity representation when their associated eigenvalues are relatively close (around 0.12 and 0.10, respectively, in Margulies et al., 2016), which is consistent with perturbation theory for eigenspace stability (Davis and Kahan, 1970). As a result, large-scale patterns such as sensory differentiation and task-related contrasts may be distributed differently across the second and third gradients.

We have included this methodological clarification in the Methods section (under the subsection “Functional and structural gradient construction”) to ensure technical clarity. Taken together, these factors likely explain the modest differences observed in FCG2 while preserving the fundamental large-scale organization of cortical connectivity.

- *Pages 3-4: two different sample sizes are reported for the pediatric cohort (5,343 vs 7,938) – please clarify.*

Response: We thank the reviewer for this comment. To avoid confusion, we have removed the imaging sample size from this sentence and now report the final analytic sample sizes clearly in the Methods section.

References

- Cole, Martin et al. (2021). “Surface-Based Connectivity Integration: an atlas-free approach to jointly study functional and structural connectivity”. In: *Human Brain Mapping* 42.11, pp. 3481–3499.
- Davis, Chandler and William Morton Kahan (1970). “The rotation of eigenvectors by a perturbation. III”. In: *SIAM Journal on Numerical Analysis* 7.1, pp. 1–46.
- Margulies, Daniel S et al. (2016). “Situating the default-mode network along a principal gradient of macroscale cortical organization”. In: *Proceedings of the National Academy of Sciences* 113.44, pp. 12574–12579.
- Nenning, Karl-Heinz et al. (2024). “Fast connectivity gradient approximation: maintaining spatially fine-grained connectivity gradients while reducing computational costs”. In: *Communications Biology* 7.1, p. 697.
- Vos de Wael, Reinder et al. (Mar. 2020). “BrainSpace: a Toolbox for the Analysis of Macroscale Gradients in Neuroimaging and Connectomics Datasets”. In: *Communications Biology* 3.1, pp. 1–10. ISSN: 2399-3642. DOI: 10.1038/s42003-020-0794-7. (Visited on 05/02/2023).

Reviewer 3

The authors have thoroughly addressed all my concerns. I specifically commend the inclusion of the HCP-Development dataset to perform harmonization. This resolves my concern regarding the comparability of data across cohorts and significantly strengthens the validity of the results. I have no further comments.

Response: We really appreciate the Referee for taking the time reviewing our paper and the positive and constructive feedback. In this revision, we have thoroughly addressed all the comments, and our point-to-point response is provided below.

Reviewer 4

Response: We sincerely thank you for your thoughtful review and valuable feedback. We have carefully addressed all the comments in the revised manuscript.